# Omecamtiv mecarbil and Mavacamten target the same myosin pocket despite opposite effects in heart contraction

Daniel Auguin [1,2,5], Julien Robert-Paganin [1,5], Stéphane Réty [3], Carlos Kikuti [1], Amandine David[1], Gabriele Theumer[4], Arndt W. Schmidt[4], Hans-Joachim Knölker [4] & Anne Houdusse [1] ✉

Inherited cardiomyopathies are common cardiac diseases worldwide, leading in the late stage to heart failure and death. The most promising treatments against these diseases are small molecules directly modulating the force produced by β-cardiac myosin, the molecular motor driving heart contraction. Omecamtiv mecarbil and Mavacamten are two such molecules that completed phase 3 clinical trials, and the inhibitor Mavacamten is now approved by the FDA. In contrast to Mavacamten, Omecamtiv mecarbil acts as an activator of cardiac contractility. Here, we reveal by X-ray crystallography that both drugs target the same pocket and stabilize a pre-stroke structural state, with only few local differences. All-atom molecular dynamics simulations reveal how these molecules produce distinct effects in motor allostery thus impacting force production in opposite way. Altogether, our results provide the framework for rational drug development for the purpose of personalized medicine.

Inherited cardiomyopathies are a global health concern, being a major cause of heart disease worldwide. Hypertrophic cardiomyopathies (HCM) are characterized by hypercontractile cardiomyocytes leading over the years to fibrosis and ventricular hypertrophy[1]. Dilated cardiomyopathies (DCM) are in contrast characterized by an hypocontractile phenotype[2,3]. While the end-stage cardiomyopathies can all lead to sudden death[4-7], the contractile phenotype differs in HCM and DCM at a molecular level[3]. These diseases are associated with single-point mutations of contractile proteins from the sarcomere, such as β-cardiac myosin[7,8]. The effect of some of these point mutations has been extensively studied but remains poorly understood. Up to now, therapeutic approaches to treat end-stage inherited cardiomyopathy have been highly invasive, including cardioverter-defibrillator implantations and heart transplantation[9].

In the heart, β-cardiac myosin is the major nanomotor that produces force during contraction. Myosins are ATP-dependent molecular motors involved in almost all processes of life (reviewed by Houdusse et al.[10]). The force produced depends in particular on the regulation of β-cardiac myosin that controls the number of active nanomotors. The double-headed β-cardiac myosin can adopt an inactive, sequestered state, unable to participate in force production, which corresponds to a specific motif called the interacting-heads motif (IHM)[11-13]. When exertion increases, destabilization of the sequestered state allows an increase in the number of myosins engaged in force production[14].

Small molecules directly targeting β-cardiac myosin can modulate force production and are promising approaches to treating cardiac diseases[15,16]. Sarcomere activators can increase contractile force[17], while inhibitors of myosin force production can decrease the force produced when the heart is hypercontractile[18,19]. Two modulators have completed phase 3 clinical trials. The activator of sarcomere contraction Omecamtiv mecarbil (OM), led to a reduction in mortality and cardiovascular events in the treatment of heart failure with reduced ejection fraction (HFrEF), (GALACTIC-HF[17]). Another modulator, the

---

[1]Structural Motility, Institut Curie, Université Paris Sciences et Lettres, Sorbonne Université, CNRS UMR144, Paris 75248, France. [2]Laboratoire de Physiologie, Ecologie et Environnement (P2E), UPRES EA 1207/USC INRAE-1328, UFR Sciences et Techniques, Université d'Orléans, Orléans, France. [3]Laboratoire de Biologie et Modélisation de la Cellule, ENS de Lyon, CNRS, UMR 5239, Inserm, U1293, Université Claude Bernard Lyon 1, Lyon, France. [4]Faculty of Chemistry, TU Dresden, Dresden, Germany. [5]These authors contributed equally: Daniel Auguin, Julien Robert-Paganin. ✉e-mail: anne.houdusse@curie.fr

inhibitor Mavacamten (Mava), became the first FDA-approved treatment, and is now available for patients of moderately severe obstructive HCM under the name of CAMZYOS™ (FDA, 2022).

These two molecules have opposite effects on the heart: while OM can increase cardiac contractility without side effects on calcium concentrations or myocardial oxygen consumption[20,21]; Mava is able to decrease cardiac contractility and suppress hypertrophy and cardiomyocyte disarray[18].

The mechanism of action of OM includes the fact that it destabilizes the sequestered state, thus increasing the number of heads engaged in force production[22]. In contrast, Mava is reported to stabilize the folded-back sequestered state[23,24] (Fig. 1a, b). However, direct measurements by FRET indicated that Mava increases the proportion of IHM by only 4% in an isolated double-headed myosin fragment (HMM)[25]. The effects of these modulators on the motor cycle, independent of their effect on the fraction of sequestered heads, must also play an important role in their mechanism to modulate heart contraction. OM increases the actin-activated ATPase activity and $P_i$ release rate but also the time spent bound to actin[26,27]. In contrast, Mava decreases the actin-activated ATPase activity, $P_i$ release rate, and affinity for actin[18,28]. We previously solved the X-ray structure of OM-bound β-cardiac myosin[29]. This revealed the allosteric binding

pocket of the drug and how it stabilizes the pre-powerstroke state (PPS), a state with a primed Lever arm, in which hydrolysis products are trapped until binding to F-actin triggers their release[29]. Interestingly, recent single-molecule data showed that OM inhibits the working stroke of the wild-type (WT) β-cardiac myosin while it increases the working stroke of the HCM mutant R712L[30,31].

Despite extensive efforts in characterizing these modulators, their precise mode of action remains unclear, including the rationale of how they can influence in the opposite way both the force output of the heart and the actin-activated $P_i$ release rate. This question could not be addressed so far since the binding site for Mava was unknown.

Structural information is also needed to design other modulators capable of enriching a personalized medicine approach. Indeed, hundreds of point mutations are responsible for inherited cardiomyopathies and these mutations have diverse effects on β-cardiac myosin function and/or regulation[32–35]. Some mutations may impede the action of modulators, either by (i) reducing their affinity or (ii) blocking their allosteric effect. For example, the efficiency of OM is reduced in the case of the DCM-causing mutation F764L[36]. It is thus essential to dissect the mode of action of OM and Mava on β-cardiac myosin, as this will guide the generation of novel drug candidates allowing all classes of mutations causing inherited cardiomyopathies to be treated.

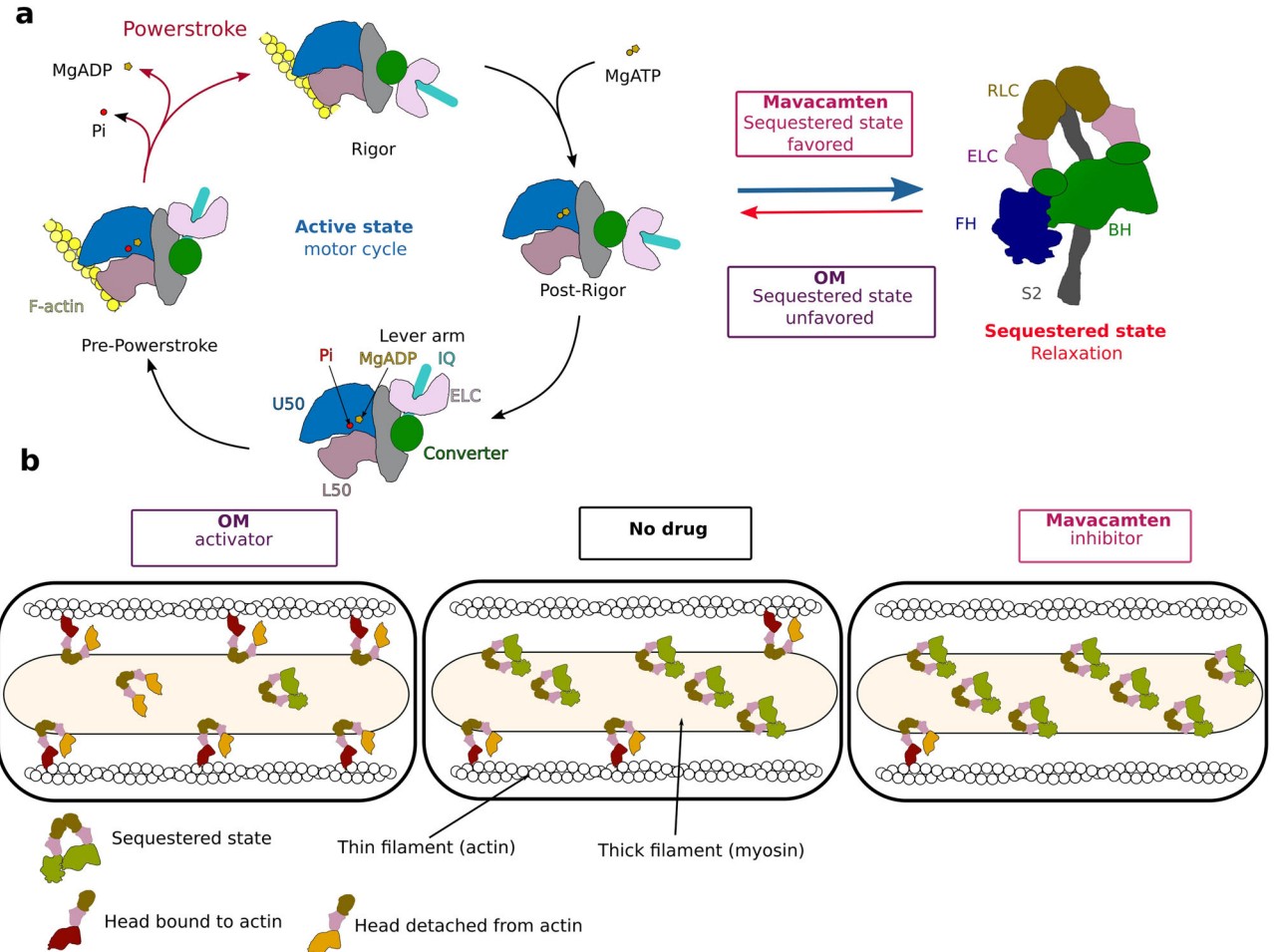

**Fig. 1 | Effect of Omecamtiv mecarbil (OM) and Mavacamten (Mava) on the motor cycle and regulation of β-cardiac myosin. a** During its motor cycle, the myosin goes through three major structural states, associating to actin in the pre-powerstroke state (PPS) with the Lever arm up and performing the power stroke which is responsible for heart contraction. In the cardiac sarcomere, the population of available myosins is highly regulated by the presence of an inactive sequestered state that shuts down a proportion of myosin to regulate force production during contraction phases. OM and Mava have opposite effects on both the motor cycle and sequestered state stability. Mava can stabilize the sequestered state while OM destabilizes it, increasing the proportion of available heads. **b** On the active population, OM stabilizes the PPS state, increasing the number of heads available to interact with actin[29], whereas Mava inhibits both the interaction with actin and the Pi release rate.

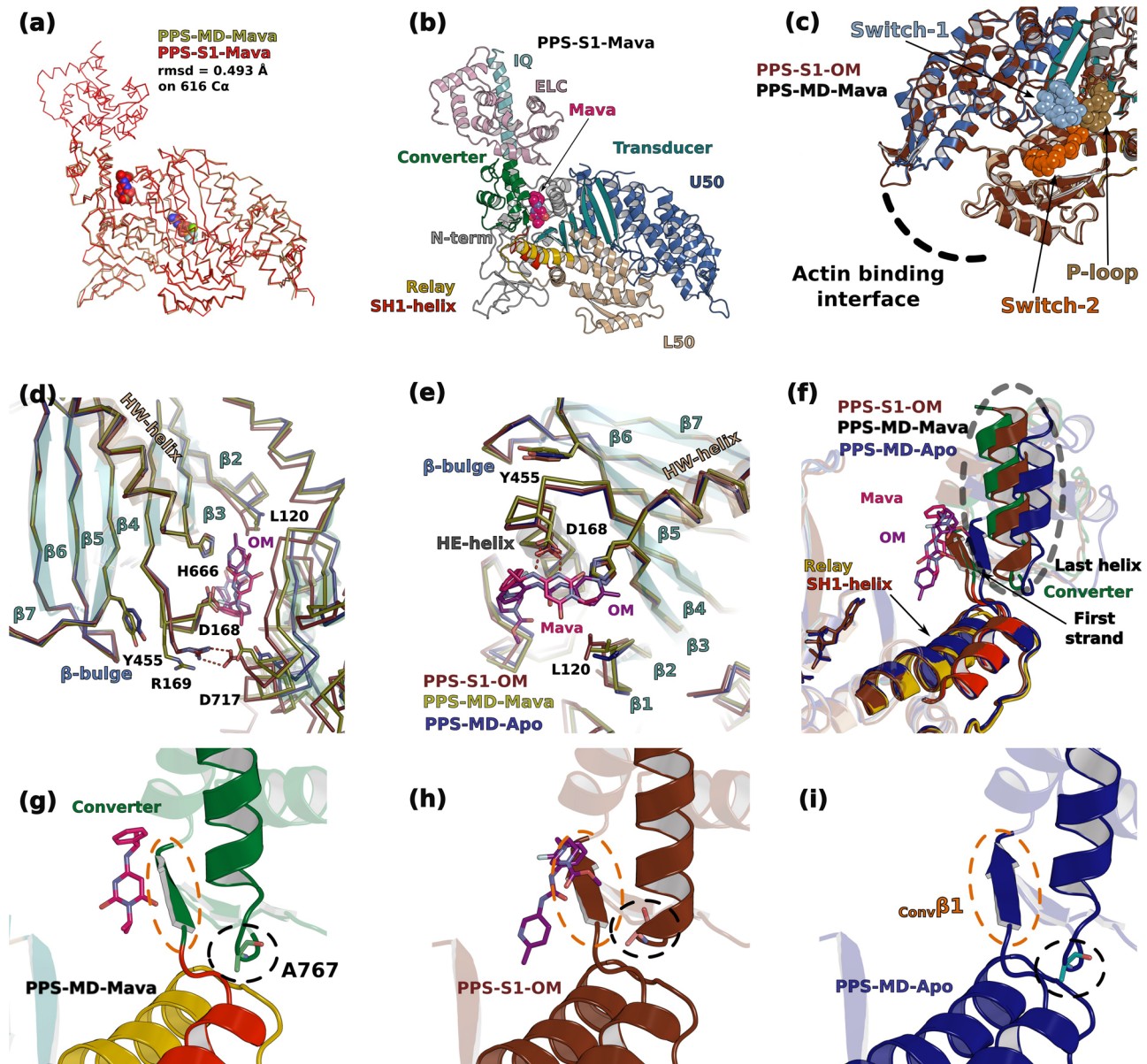

**Fig. 2 | Crystal structures of β-cardiac myosin complexed with Mavacamten and comparison with other PPS structures. a** The motor domain (MD) and the S1 fragment (S1) structures complexed to Mavacamten (Mava) superimpose with a rmsd of 0.493 Å on 616 Cα. **b** Overall structure of the S1 fragment of β-cardiac myosin bound to Mava. The compound occupies a pocket between the Lever arm and the motor domain. **c** Superimposition of the OM bound and the Mava bound structures on the U50 shows that there is no difference of conformation on the actin-binding interface and on the three nucleotide-binding loops of the active site: Switches-1 and −2 and P-loop (spheres). **d–f** Local conformational differences induced by the presence of OM or Mava, notably in linkers near the Transducer and in the Lever arm orientation. The Lever arm is more primed in the structures in which either of these drugs are bound, compared to the Apo structure. **g–i** Conformation of Ala767 and of the first β-strand of the Converter (<sup>Conv</sup>β1) depending on the presence of drugs.

In this work, we solved the structure of Mava-bound bovine β-cardiac myosin. We found that the allosteric binding pocket of this inhibitor is the same as that of OM, unexpectedly. All-atom molecular dynamics simulations explain the source of their opposite effect on myosin allostery. This provides the blueprint of the mechanisms underlying the modulation of myosin activity and regulation through the OM/Mava pocket.

## Results

### Mavacamten stabilizes a pre-powerstroke state
We solved the structures of (i) the S1 fragment (3-807 in chain A and 3-798 in chain B of the asymmetric unit) complexed with Mava and Mg.ADP.BeFx (PPS-S1-Mava), (ii) the proteolyzed motor domain (MD, 34-781) complexed with Mava and Mg.ADP.BeFx (PPS-MD-Mava) and

(iii) the MD fragment (32-780) complexed to Mg.ADP.Vanadate (PPS-MD-Apo) at a resolution of 2.61; 1.80 and 2.76 Å respectively (Supplementary Tables 1 & 2). Reprocessing of the OM-bound dataset[29] with the StarAniso procedure[37] provided a 1.96 Å resolution structure (PPS-S1-OM). The drugs were identified without ambiguity in the electron density map (Supplementary Fig. 1). Despite the different space groups and environments, the Mava-bound structures are nearly identical (Fig. 2a), which is a strong validation of the binding site. Like OM, Mava co-crystallizes with cardiac myosin in a pre-powerstroke (PPS) state, a state that traps the hydrolysis products with a primed Lever arm. Visualizing Mava and OM bound in the PPS state is relevant and consistent with the transient kinetic studies[26–28], showing that these drugs are compatible with ATP hydrolysis, yet lead to slow $P_i$ release rate in the absence of actin (and thus slow basal rate). Upon drug binding, the

major difference found is in the Converter position (Fig. 2a and Supplementary Fig. 2).

## OM and Mava target the same pocket

Unexpectedly, both drugs target the same pocket, located between the N-terminal (N-term) and the Converter subdomains of the motor domain (Fig. 2b). No differences are found in the active site, including in the Switch-2 position (Fig. 2c). These structures are also similar for the conformation of the internal pocket – so called 50 kDa cleft – whose closure/opening controls the affinity for the actin filament (Fig. 2c). Such a similarity between the OM- and the Mava-bound structures is unexpected since OM and Mava strikingly differ in the way they increase (OM) or reduce (Mava) the rates of actin-activated $P_i$ release[26–28]. A close comparison of the three PPS structures reveals that the differences are small and exist mainly around the drug binding site: (i) local differences in the $^{N-term}$HE-helix and the preceding linker (both involved in drug binding) (see D168 and R169, Fig. 2d, e), extend to small differences in the $^{Transducer}$β4 strand (see Y455 in Fig. 2d, e), near the $^{Transducer}$HO-linker and the $^{Transducer}$β-bulge (ii) nonidentical interactions of the drugs with the Converter (Fig. 2d, f and Supplementary Movie 1) also lead to local differences in particular for the position of $^{Conv}$Ala767 at the beginning of the last helix of the Converter (Fig. 2g–i). OM stabilizes the first turn of this helix but not Mava (Supplementary Movie 1). In the two drug-bound structures, closure of the allosteric pocket around the drug results in a similar priming of the Lever arm angle for the two conditions (~78° for OM and Mava, versus ~69° for Apo) (Supplementary Fig. 3a–f and Supplementary Movie 1). Differences in drug interactions with the Converter also translate to differences in the Relay, SH1-helix, and Converter orientations (Supplementary Fig. 4), while distinct interactions of $^{HW}$H666 with both drugs lead to small effects on the first two Transducer strands (β1 and β2, near L120, Fig. 2d). The end of the $^{Transducer}$β2 strand (L120) uniquely participates in Mava binding. In addition, the size and conformation of the drug-binding pocket differ between OM, Mava, and the Apo structures (Supplementary Fig. 3g–l). Only in the most closed configuration found when OM binds, does the beginning of the β4 strand (R169) interact with $^{Conv}$D717 (Fig. 2d). Thus, significant distinctions in the drug interactions affect both the Transducer and the Lever arm, which are key elements of the allosteric communication in myosin motors, involved in force generation[10].

## Interactions mediated by OM and Mava in the crystal structures

To compare how Mava and OM bind in this pocket, the drug moieties were delineated and named A to C for Mava and A to D for OM (Supplementary Fig. 5). The drugs mediate similar polar bonds with critical residues of the two pockets (Fig. 3a, b and Supplementary Table 3): direct interactions involve $^{N-term}$Asp168, the amide nitrogen of the $^{Conv}$Arg712 backbone and the side chain of $^{Conv}$Asn711 with the carbamoylamino group of OM, and the amino and pyrimidine groups of Mava. Mava and OM are also involved in similar apolar interactions (Fig. 3a, b and Supplementary Table 3). A few additional residues differ between the OM and Mava pockets since OM is thinner but longer than Mava (Fig. 3a, b). The isopropyl group of Mava uniquely interacts with $^{N-term}$Leu120 (Figs. 2d and 3a). In contrast, the methyl ester piperazine ring of OM uniquely reaches residues of the linker prior to the HE helix as well as the last helix of the Converter on one end (K146, R147, $^{HE}$N160, $^{Conv-H3}$A767), while the methyl-pyridine reaches the Relay (H492, E497) on the other end of the OM molecule (Fig. 3a, b).

Sequence differences among different myosins-2 for residues around this pocket correlate with drug specificity. As OM, Mava is specific for α- and β-cardiac myosin. Fast skeletal myosin is inhibited with a ten-fold higher $IC_{50}$, while smooth and non-muscle myosin-2 are not inhibited[28]. The weaker activity for fast skeletal myosin comes from only two small sequence differences in residues that directly bind the drugs ($^{Card}$Y164/$^{Sk}$F, $^{Card}$N711/$^{Sk}$S) (Fig. 3c and Supplementary Fig. 6).

In contrast, several sequence differences are responsible for the inability of Mava or OM to target other class-2 myosins (Fig. 3c).

## Mobility of the primed Lever arm greatly differs when Mava or OM are bound

To decipher how small local differences near the binding site of these force modulators can translate into different energy landscapes and thus different control of actin-binding or $P_i$ release from the motor, all-atom molecular dynamics simulations were performed. Such simulations provide indications for allosteric communication and subdomain reorganization when conformational sub-states are explored[34,38–43]. We made a comparative study between three simulations starting from S1 bound to Mava, S1 bound to OM, and S1 without compound (Apo), all bound to Mg.ADP.$P_i$ in the active site (Supplementary Movie 2, 3, and 4). To guarantee the reproducibility of the results and minimize artifacts, the simulations were performed at least twice, from independent minimizations of the structures (See Methods).

We monitored the Lever arm position by plotting the root-mean-square displacement (RMSD) for backbone Cα atoms relative to their initial minimized complex structures during each simulation (Cα RMSD plot, Fig. 4a). In addition, we computed its orientation in two planes (Fig. 4b, c). In the Apo condition, the Lever arm swings back and forth, exploring conformations differing by up to ~36° (Fig. 4a, Supplementary Figs. 7, 8 and Supplementary Movie 2). All our indicators converged, as the RMSD curves follow a similar evolution to the orientations in the two planes (Fig. 4a–c). The Lever arm is the most mobile in the Apo condition, with large movements leading to explore less primed Lever arm states. Mava slightly restrains the amplitude of the movements (Supplementary Figs. 8, 9). In contrast, OM binding significantly restrains the Lever arm movements and maintains it highly primed (Fig. 4a–c and Supplementary Figs. 8, 10). This could be demonstrated by statistical approaches such as expressing the RMSD function of the radius of gyration (Rg) of the entire protein, which can be plotted with the density probability function of the frames (Fig. 4d–f) or with computed Gibbs free energy depending on the occurrence of the conformations during the simulation (Fig. 4g–i). The presence of OM and Mava both exert a cohesive action between the Lever arm and the motor domain that contributes to maintaining the Lever arm primed (Supplementary Movies 2, 3, and 4). Interestingly, OM and Mava significantly differ in their mobility within the pocket: OM remains in a specific position for a longer duration (close to that found in the crystal structure, Supplementary Fig. 10 and Supplementary Movie 5). In contrast, Mava explores different positions in the pocket (Supplementary Fig. 9 and Supplementary Movie 6). This likely explains the distinct Lever arm mobility observed in the two conditions (Fig. 4d–i). Thus, distinct fluctuations when Mava or OM is bound in the pocket correspond to distinct possible movements of the Converter/Lever arm orientation and position explored (Supplementary Movie 7).

## Distinct effects of Mava and OM on motor domain allostery

Since the actin-dependent $P_i$ release rate is activated by OM and decreased by Mava, we then analyzed the dynamics within the motor domain, and in particular how the relative position of nucleotide-binding elements (P-loop, Switch-1 and Switch-2) behave along the simulations. These elements constitute the "backdoor", which is closed in the pre-powerstroke state, but must open during actin-activated $P_i$ release[10,44]. The dynamics within the motor domain are of lower amplitude compared to that found for the Lever arm, as shown by the Cα RMSD plot (Fig. 5a). Interestingly, while the Apo and OM conditions follow similar dynamics, Mava is the most agitated and its dynamics significantly differs (Fig. 5a). In order to monitor the stability of the actin-binding interface that involves elements from the U50 and the L50, we followed the relative dynamics of the Helix-Turn-Helix (HTH) (Fig. 5b) and of the HO-helix in each condition (Fig. 5c). The HTH is part

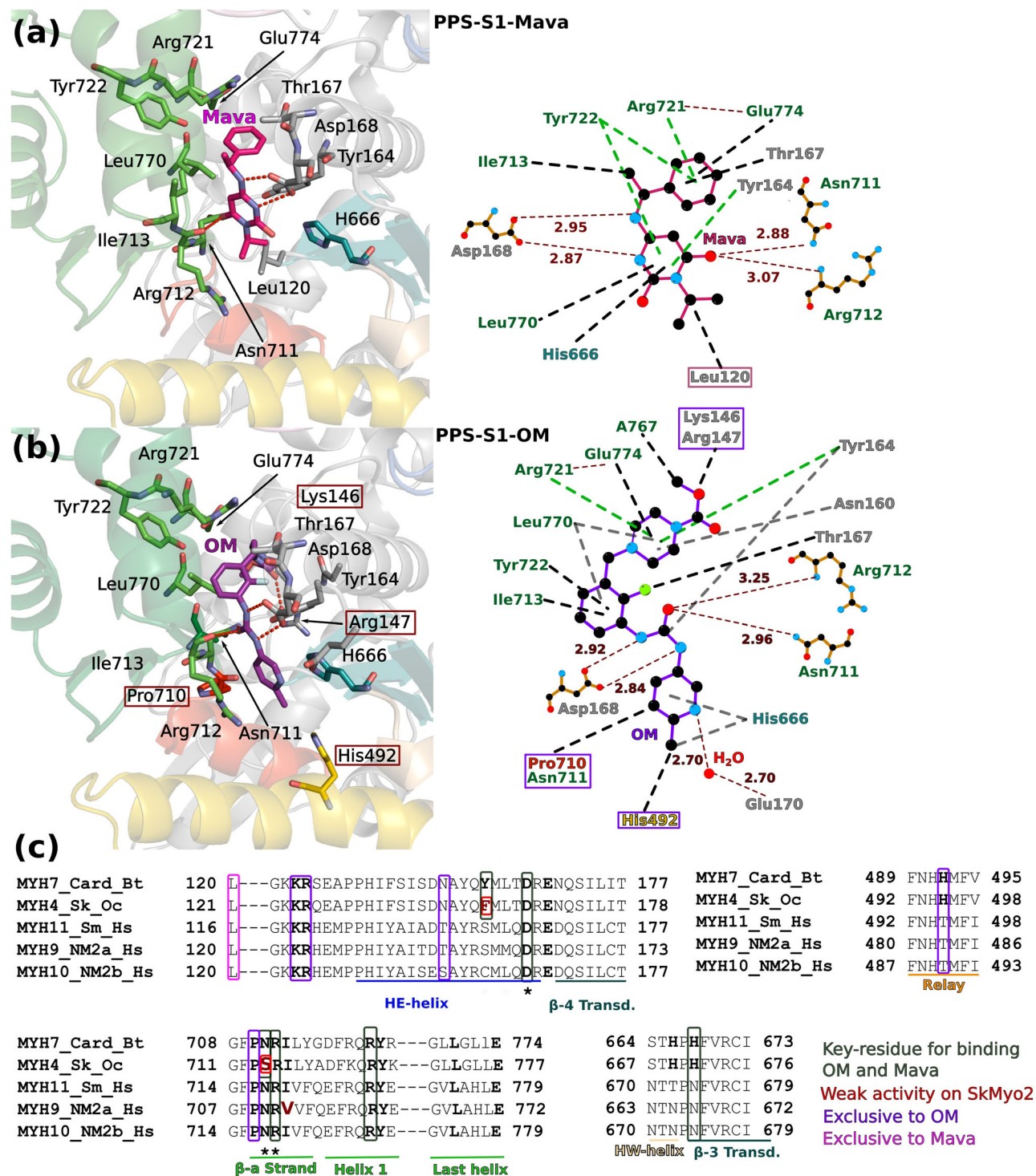

**Fig. 3 | Mava and OM target the same pocket. a** Cartoon and schematic representation of the Mava binding site. Contact distances are indicated in Å. **b** Cartoon and schematic LigPlot-based[71] representation of OM binding site. **c** Sequence alignment of class 2 myosins highlights the specificity determinants of OM and Mava. Key-residues involved in binding both drugs are boxed in green, residues that lead to the weaker affinity for skeletal myosin are colored in red, and residues exclusive to OM or Mava binding are boxed in deep purple and pink respectively. A star below the sequence indicates residues involved in electrostatic interactions.

of the actin-binding interface and an element of the L50 and the position fluctuation of the long HO-helix of the U50 can be a reporter of the dynamics and position of this subdomain.

The dynamics of the connectors in the Apo condition reveal that the opening of the backdoor is explored via a Switch-2 movement that leads to the loss of the salt bridge between Switch-1 and Switch-2 (R243/E466) (Fig. 5b, Supplementary Fig. 7 and Movie 2). These

fluctuations represent the first exploration of a transition towards the $P_i$ release state, without much change in the L50 subdomain conformation and only small changes for the outer internal pocket (Fig. 5b, c and Movie 2). In the OM simulation, the backdoor displays high dynamics but does not fully open, as it is the case in the Apo condition (Fig. 5d, e, Supplementary Fig. 10a–c, Movies 2, 3). For Apo and OM, the communication between Switch-2 and the $^{L50}$Wedge is

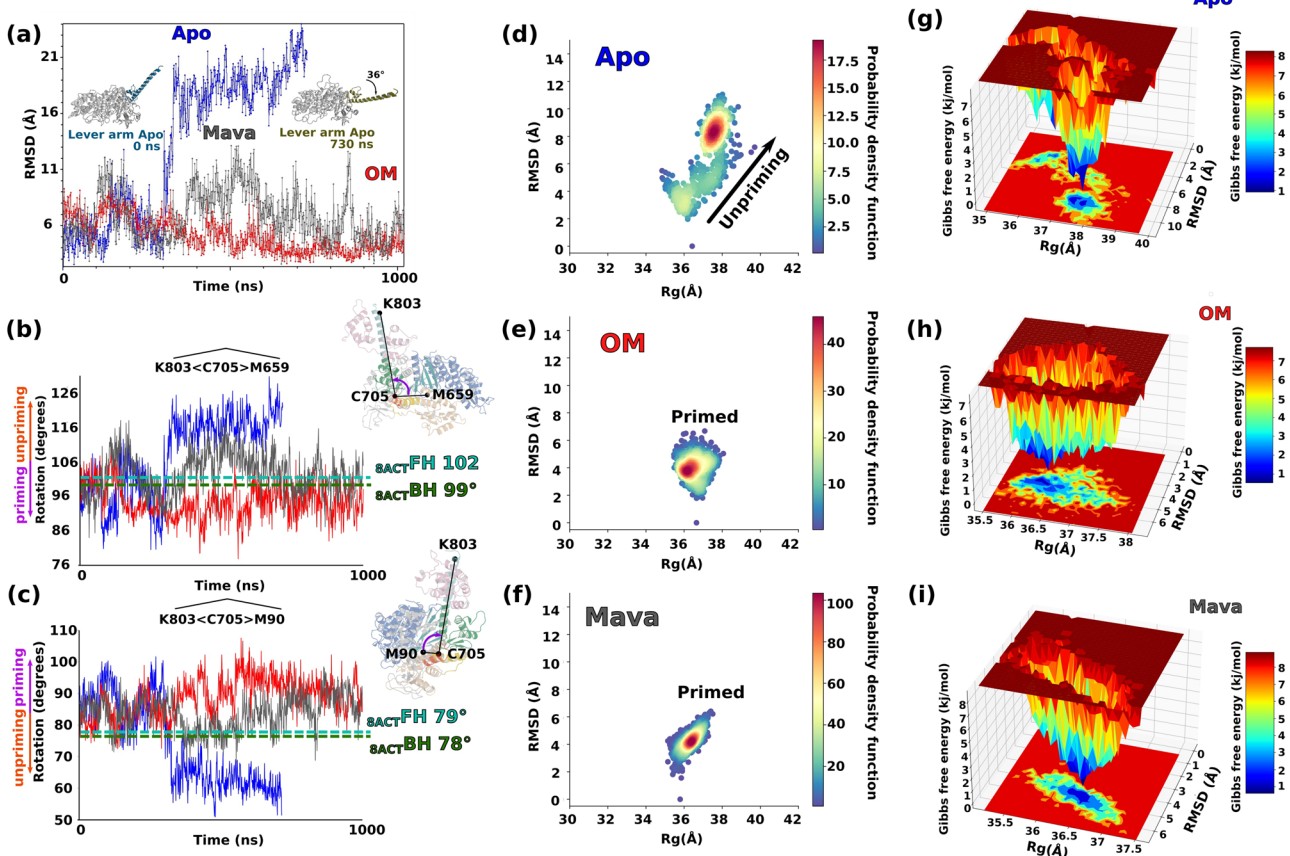

**Fig. 4 | OM and Mava stabilize the priming of the Lever arm. a** Evolution of the root mean square deviation (RMSD) of the Cα positions of the Lever arm (711-806) for the three conditions of the simulation: Apo (blue), OM (red), and Mava (stone gray). The two extreme positions of the Apo (0 ns and 730 ns) are represented in blue and sand green, showing that a 36° swing occurred during the simulation. In the presence of OM or Mava, the lever arm remains primed with fluctuations. **b, c** monitors the evolution of the Lever arm orientation for the three simulations via two distinct angles, using K803 (tip of the Lever arm), and C705 (base of the Lever arm). The angles found for the blocked head (BH, green dashed lines) and of the free head (FH, light blue dashed lines) of the cryo-EM β-cardiac IHM structure (PDB code 8ACT[11]) are indicated. Since the Lever arm is kinked in the BH, the measurement was not performed with K803 but with L781, which is the last residue before the kink. **d-f** display the RMSD function of the radius of gyration (Rg) calculated on the entire protein on the Apo, OM, and Mava conditions respectively. A probability density function allows to monitoring the most populated conformations along the simulation. The Rg can be used to observe a change in the Lever arm since the unpriming of the lever arm triggers an increase in the Rg (see (**d**)). **g-i** show energy landscapes for all the conditions. Gibb's free energy is plotted against Rg and the RMSD.

maintained via F465 (Fig. 5d, e), which restrains the L50 subdomain orientations in a PPS-like position. The overall landscape of states explored during the simulations of both Apo and OM describe small-range fluctuations at the actin interface that are coupled with a dynamic backdoor that keeps $P_i$ trapped. These data are consistent with an exploration of states along the OM simulation that are similar to those of Apo, as they do not deviate much from the initial PPS conformation, with minimal changes in the actin interface since the position of the HTH and the HO-helix remains stable during all the time course of the dynamics (Fig. 5b, c). Apo and OM differ mainly in the fact that the Lever arm dynamics are quite reduced when OM is bound. Since OM allows fast rates of actin-driven $P_i$ release, these simulations suggest that the actin-driven transitions that drive $P_i$ release can occur without a large Lever arm swing.

In contrast, the simulations with Mava bound show that movement of the Switch elements around the $P_i$ occurs on the first ns of the simulation, prior to a movement of Mava from its initial position. This movement closes the backdoor, which remains in this closed position throughout the simulation thanks to electrostatic bond compensation (Fig. 5f, Supplementary Fig. 9, and Supplementary Movie 4). After rotation of the F465 side chain (at 11 ns), the communication between the Switch-2 and the Wedge is quickly lost around 36 ns (Fig. 5f,

Supplementary Fig. 9, and Supplementary Movie 4). This loss of anchoring induces a large divergence in the L50 orientation compared to what is explored when OM or no drug are bound. The L50 subdomain behaves as if it is no longer maintained while Switch-2 remains in a fully closed position, trapping $P_i$ (Fig. 5b, c, f and Supplementary Movies 2, 3, 4). Close analysis of the simulations suggests that specific interactions of Mava with $^{HW}$H666 and with the β1-β2 strands of the Transducer could create distinct constraints, central for altered communication between the drug-binding site and the active site (Supplementary Fig. 9). Thus, the simulations demonstrate that the presence of Mava alters the energy landscape of myosin conformations explored. The new allosteric communication within the head not only maintains $P_i$ trapped but also translates into divergent movements of the outer part of the internal myosin cleft. Increased freedom for the L50 subdomain significantly alters the actin-binding interface, which deviates from the PPS-like conformations explored when OM is bound, or no drug is present (Apo). This is confirmed by the relative dynamics of the HTH compared to the HO-helix (Fig. 5b, c). While the HO-helix remains stable, the position of the HTH rapidly deviates from its position, indicating a strong alteration of the actin-binding interface. Consequently, Mava-bound substrates are inefficient starting points of the powerstroke (Fig. 5f).

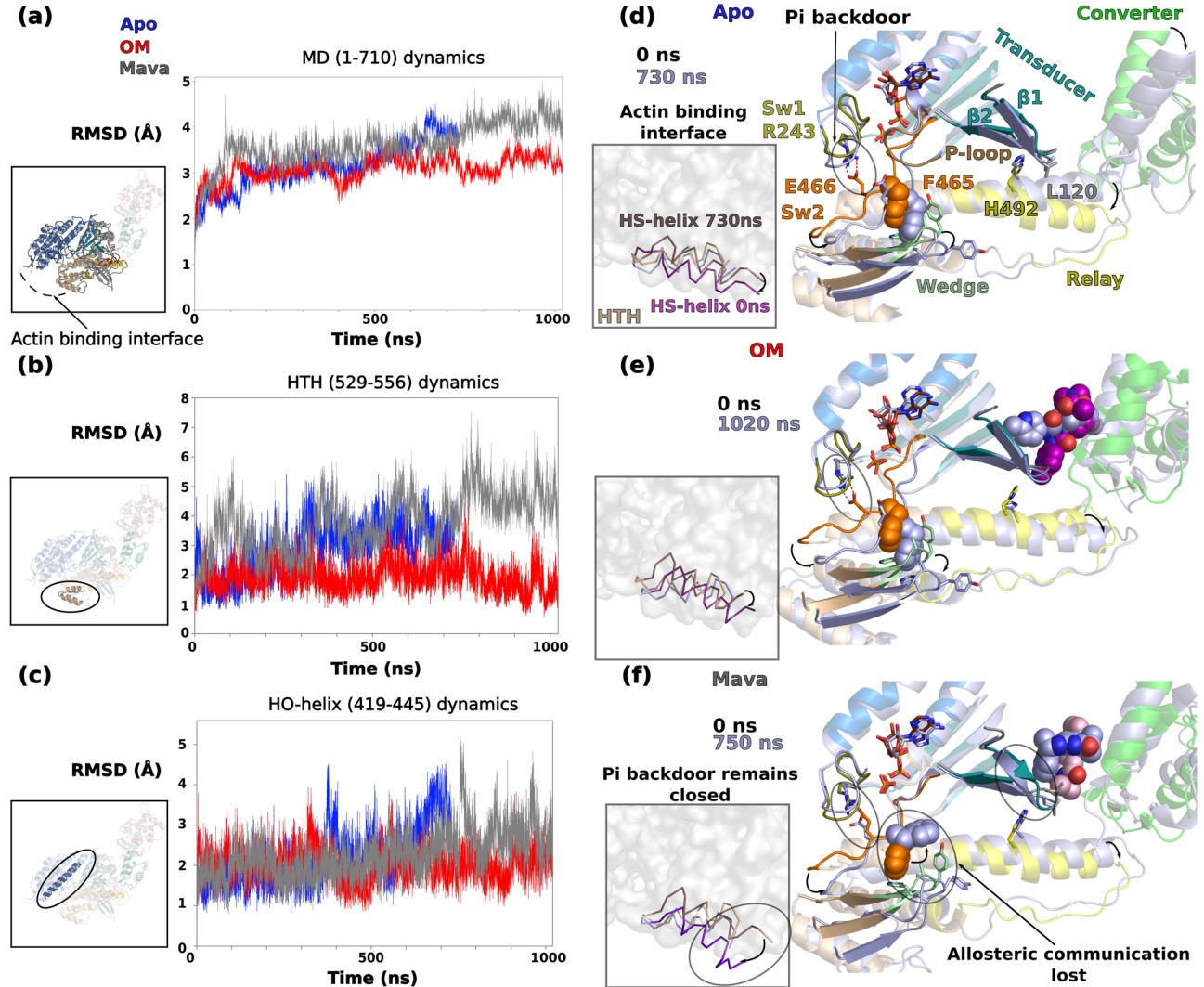

**Fig. 5 | OM and Mava induce different effects on the allostery of the motor.**
**a** RMSD of the Cα positions of the motor domain (1-710) during the time course of
the simulation for the three conditions: Apo (blue), OM (red), and Mava (gray).
Following the same methodology, (**b**) and (**c**) RMSD of the Cα positions of the helix-
turn-helix (HTH, 529-526) and of the HO-helix (419-445), respectively. In (**d**–**f**), the
extreme positions of the HTH (part of the actin-binding interface, the HS-helix is
colored differently) are represented as well as that of the connectors for Apo, OM,
and Mava, respectively. The $P_i$ backdoor is found close to the salt bridge between
Switch-1 (Sw1) R243 and Switch-2 (Sw2) E466. In both the Apo and the OM condi-
tions, the communication between the Wedge and the Sw2 remains during the
duration of the simulation through the interaction with $^{Sw2}$F465. This communica-
tion stabilizes the conformation of the HTH and of the actin-binding interface. In
the Mava condition, the communication between the Wedge and Switch-2 is rapidly
lost in the simulation with the flip of F465, leading to a drastic change in the
conformation of the HTH and specifically the HS-helix (**f**). When Mava is bound, the
$P_i$ backdoor remains closed during the entire simulation.

In addition, we observe rapid pocket exploration by the small
Mava molecule, which can lead to its release in the solvent in one of the
simulations, and allows Mava to occupy several internal pockets that
are closed in other simulations. Mava indeed takes opportunities of
movements of the β1-β2 strands and the linker preceding the HE helix
to fit in distinct pockets (Supplementary Fig. 9 and Supplemen-
tary Movies 4, 6, 7), thus stabilizing atypical conformations of the
motor inadequate for Pi release or actin binding. This strongly differs
from the much-reduced dynamics observed for the longer OM mole-
cule that fits in a closed drug-binding pocket stabilized by polar
interactions made between residues of the N-term and Converter
subdomains (Supplementary Fig. 10, Supplementary Movies 2, 3, 5, 7).
These differences in the mobility of Mava compared to OM are
reflected in the $K_D$ calculations over the time course of the simulation
(Supplementary Fig. 11). While the calculated $K_D$ of OM oscillates
around 1 μM (average $K_D$ 1.30 μM) and remains stable, that of Mava
goes through larger oscillations due to the occupation of diverse
pockets (average $K_D$ 3.33 μM).

## A proposed mechanism for Mava
The structures combined with the dynamics allow to propose a
mechanism of action explaining how two drugs that bind to the same
pocket can have distinct effects. The two drugs maintain the Lever
arm up but have (i) different dynamics within the pocket and (ii)
distinct allosteric effects due to their differences in shape and
structure.

The small and bulky Mava fits in a slightly wider pocket compared
to the elongated OM. It maintains the Converter further away from the
motor domain. Since it is only composed of two cyclic groups, Mava
does not establish interactions with the Relay, yet it explores pockets
that are found near the first three strands of the Transducer (Supple-
mentary Fig. 9 and Supplementary Movie 4, 6, 7). Thus, it does not
exert restraints on the Relay, explaining why the position of this con-
nector deviates during the simulation, allowing both the Wedge and
Switch-2 to lose communication and reach uncanonical positions. This
has two consequences: **(i)** the backdoor maintains a closed con-
formation as Switch-2 movements greatly differ from what is observed

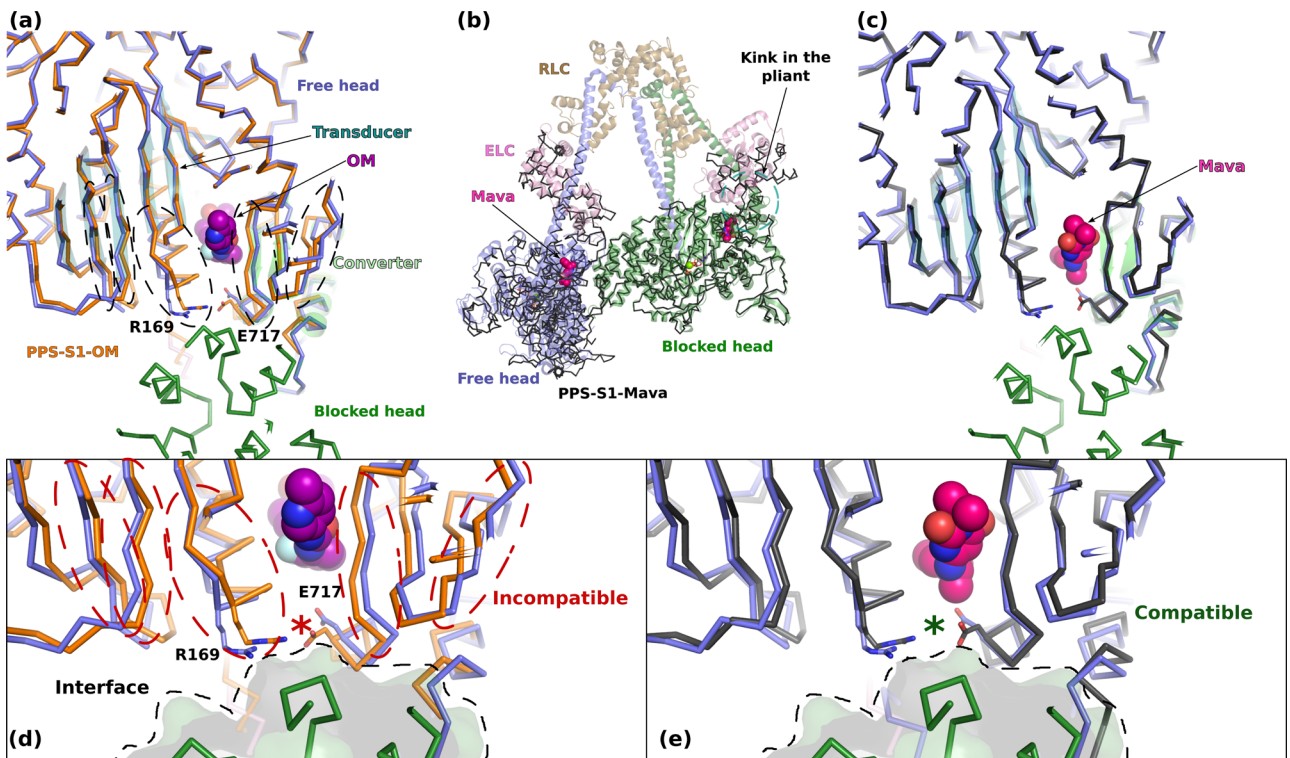

**Fig. 6 | Effects of Mava and OM on the β-cardiac myosin sequestered state.**
**a** Superimposition of the cryo-EM β-cardiac IHM structure (cartoon, PDB code 8ACT[11] on the S1 structure of β-cardiac myosin complexed to OM (PDB 5N69[29]), Superimposition on the motor domain of the blocked head (BH) (residues 1–707). The presence of OM induces several differences in the Transducer and in the Converter (indicated in dashed lines). **b** Superimposition of the β-cardiac IHM structure on the two molecules of the asymmetric unit of PPS-S1-Mava (ribbon, in black) superimposed on the head-head interface (residues 329–447 of the BH and residues 498–518, 708–780 on the FH, RMSD 0.5 Å on 112 Cα). **c** Superimposition of a molecule of the asymmetric unit of PPS-S1-Mava on the BH (residue 1–707) of the β-cardiac myosin IHM. **d** Zoom at the head-head interface of the superimposition of PPS-S1-OM and the FH of the sequestered state. The FH is represented as a transparent surface to highlight the surface of interaction. The differences between PPS-S1-OM and the FH making the presence of OM incompatible with the IHM are indicated in red dashed lines. These differences also disrupt the electrostatic interaction between R169 and E717 (indicated with a red star) that is present in the head-head interface of the IHM. **e** Zoom at the head-head interface of the superimposition of PPS-S1-Mava and the FH of the sequestered state. The structures are highly similar and the presence of Mava does not disrupt the interaction between R169 and E717. The binding of Mava is thus compatible with the IHM.

for Apo myosin; (ii) the actin-binding interface is altered. Mava binding thus results in "incompetent PPS" states that interact poorly with actin and thus cannot release $P_i$ readily. This is fully consistent with in vitro data demonstrating that when Mava is bound to β-cardiac myosin, the $P_i$ release rate is slowed with and without actin and the affinity of the myosin head for actin is reduced[18,28].

Conversely, OM is elongated and able to interact with the Relay helix. It consequently not only stabilizes the Lever arm up, but also exerts restraints on the Relay position. In contrast to Mava, it favors states with a primed Lever arm that are also able to interact with actin readily (Supplementary Fig. 9 and Movies 3, 5, 7).

**Effects on the formation of the sequestered state**
Since reports indicate that OM destabilizes the sequestered state while Mava seems to stabilize it[22,23,25], we examined whether the high-resolution structures reported here were providing insights on this potential opposite effect. In cardiac IHM, both heads adopt a PPS state[11], a structural state favored when ADP.Pi is bound, and which is largely similar to those favored by binding of OM or Mava. We next compared the orientation of the Lever arm during the time course of molecular dynamics simulations in the presence of OM or Mava to the angles adopted by both the blocked head (BH) and the free head (FH) of the IHM[11] (Fig. 4b, c). Interestingly, OM clearly induces priming of the Lever arm during all the simulations, which would not be compatible with the formation of the IHM. In contrast, Mava allows a priming of the Lever arm that might be compatible with those required to form the IHM. These small differences in the Lever arm orientation could be

the first reason for the opposite effects of the two compounds on the sequestered state stability.

We finally examined the precise Transducer conformations as well as their relative positioning to that of the Converter. The surface of these two structural elements of the motor domain indeed constitutes the FH mesa – a flat region recognized by the BH[11,14]. Importantly, the modulator bound in the OM/Mava pocket plays a key role in the relative positioning of the Transducer and the Converter. We demonstrate that when OM is bound, the surface differs greatly from that required for forming the FH Mesa due to the narrow size of the pocket and the change in the Transducer conformation (Fig. 6a). In contrast, the same surface of the Mava-bound structure does not differ significantly compared to that found in the Apo structure and could thus allow interactions found between heads in the IHM (Fig. 6b, c). Quite interestingly, in the PPS-S1-Mava structure, the two molecules in the asymmetric unit interact with each other by forming interactions that are similar to those found between heads in the IHM (Fig. 6b). In summary, the Mava conformation does not differ from that required to form intramolecular IHM interactions mediated by the BH or the FH heads[11] (Fig. 6c). In contrast, the OM-bound structure indicates that the activator changes the conformation of the Transducer, providing an additional clue to explain why OM destabilizes the IHM (Fig. 6d, e).

## Discussion
Small molecules able to modulate the force produced by myosins are today the best hope for treatments against acute heart failure and inherited heart diseases such as cardiomyopathies[45]. Amongst these

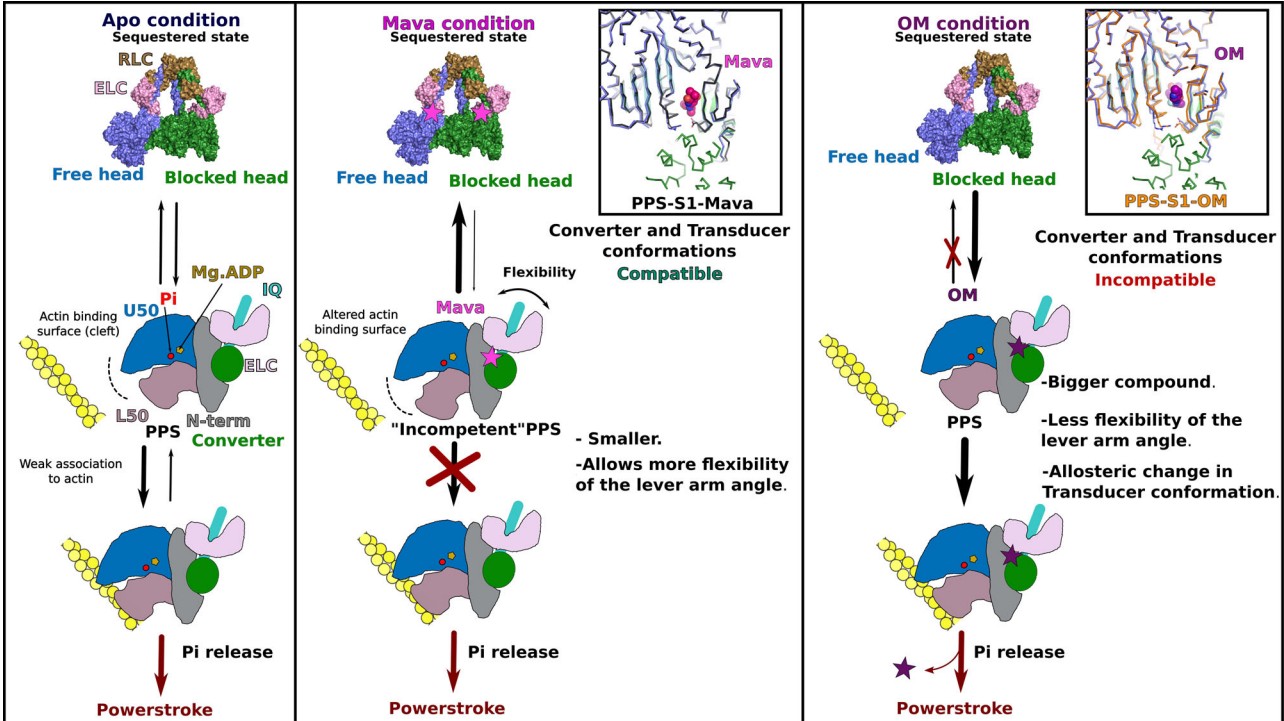

**Fig. 7 | OM and Mava mechanism of action.** Schematic representation of the effects of both OM and Mava on (i) folding back to form the sequestered state and (ii) $P_i$ release. (Left) The Apo condition (without drug) allows the dynamic formation of the sequestered state (Interacting-heads motif, IHM) in equilibrium with heads available for association with actin. (Center) Mava alters the actin-binding interface, generating an "incompetent" PPS that slows association with the actin filament. Mava-bound heads undergo flexibility in Lever arm priming, a requirement for the formation of the head/head interfaces characteristic of the cardiac sequestered state[11]. In the zoom (inset), the cardiac IHM blocked and free head structures are compared with the Mava-bound crystal structure (thin black lines). (Right) When OM binds (thin orange lines), the conformation of the Transducer and the orientation of the Lever arm differ from what is found in cardiac IHM. PPS-OM is thus incompatible with the sequestered state. OM bound heads can efficiently bind F-actin and promote the $P_i$ release, yet the swing of the Lever arm is affected as it requires destabilization of the OM binding site[30].

treatments, the activator Omecamtiv mecarbil (OM) is in phase 3 clinical trials against heart failure[17], and Mavacamten (Mava) was approved by the FDA under the name of CAMZYOS™ to treat hypertrophic cardiomyopathies[46]. As such small molecules might be used as treatments, their precise mechanism of action must be revealed. This requires an understanding of how they allosterically influence motor function. In addition, this information is of invaluable interest for the conception of specific myosin modulators that can treat a large range of myosin-related diseases[15].

In this work, we combined high-resolution X-ray structures and molecular dynamics to elucidate the molecular mechanisms of OM and Mava, two molecules with opposite effects on the force output of the heart[18,21,28]. We reveal that Mava occupies the same pocket as OM, a binding site that only forms in states of the motor in which the Lever arm is primed. Local changes near the drug-binding site differ when the Apo, OM, and Mava structures are compared and these small differences can be transmitted to the Transducer and the Lever arm which are key elements of the allosteric communication in myosin motors[10].

Detailed structural analysis coupled to molecular dynamics reveals how Mava triggers exploration of "incompetent" PPS states in which the actin interface and coupling within the motor are perturbed, consistent with the previously observed decrease of $P_i$ release rate[28]. In contrast, OM favors the exploration of structural states that are compatible with the opening of the back-door while keeping the Lever arm primed, states that are likely favoring inducing efficient binding to F-actin, as previously proposed from structures of the Pi release states of Myo6[44]. Single-molecule experiments demonstrated that OM is compatible with actin binding, although it maintains the Lever arm primed, acting as a suppressor of the powerstroke[30]. Our results are fully consistent with this data, as the cohesive action of OM around its

binding site can help maintain the Lever arm primed with limited flexibility when it occupies the pocket. This can be explained by the elongated shape of OM that allows for example interactions of the compound with the Converter and the Relay, not possible for Mava. These interactions stabilize the position of the lever arm. In contrast, Mava does not maintain the Lever arm in such a specific primed position, it slows actin binding due to the fact that the actin-binding interfaces of the motor explored when Mava is bound differ from those found when Mava is not bound, since Mava uncouples the myosin subdomains. This particularly affects the internal cleft, whose conformation and dynamics greatly impact the actin-binding interface. Interestingly, we observed that the closure of Switch-2 could also participate in reducing the $P_i$ release rate[18,28] by closing the backdoor. In light of the structural states explored, while OM increases the number of competent heads able to interact with actin filament and produce force, Mava reduces this number by trapping the heads in an incompetent ensemble of states that cannot efficiently interact with actin and initiate force production (Fig. 7). Thus, all-atom dynamics simulations provide essential insights on how binding of small modulators alter the allosteric communication within a myosin motor, in particular by altering the conformational ensembles explored by the motor domain and decoupling these movements from the priming of the Lever arm.

Mava and OM have also been reported to have opposite effects on the sequestered auto-inhibited state of cardiac myosin[22-25]. Thus, the two drugs also affect the ability of two heads to form the IHM interface, although they bind in the same pocket. The high-resolution structure of β-cardiac myosin IHM without drug bound[11] provides a precise description of the head/head interfaces. The FH Mesa corresponds to the surface of the FH head involved in interactions with the BH head, it involves the Transducer and the Converter. The OM structure indicates

why the surface required to bind to the BH head is not compatible with the formation of the IHM. Closure of the pocket around OM and allosteric changes on the Transducer both impose important changes in the FH Mesa that can prevent the formation of the IHM (Fig. 7). In the Mava-bound structures, no allosteric alteration of the Transducer conformation is observed, and all-atom simulations indicate that the mobility of the drug binding pocket is enough to allow the interactions required for IHM stabilization. Consistently, FRET measurements have indicated that Mava increases the number of heads forming the IHM by ~4%[25]. Interestingly, this amount is one order of magnitude less than previously reported when a crosslinked version of Mava has been used[23]. Currently, a high-resolution structure of the IHM with Mava bound is lacking. The medium-resolution structures (~6 Å at most[12,13]) cannot directly assess whether the modulators are bound. Given the effects of Mava on the motor domain conformation observed in dynamics, we cannot exclude that the drug induces differences in the IHM conformation that are not visualized in the filament structures due to the resolution limits. Future studies of a high-resolution Mava-bound cardiac IHM are thus required. Importantly, the effects of Mava on the stabilization of the IHM seem modest[25]. Its 10-fold higher effect on mant-nucleotide exchange – often called SRX measurements[25] –indicates that the drug efficiently inhibits product release. Uncoupling within the motor head as seen in our all-atom simulations provides insights on why the presence of Mava induces slower binding to F-actin, as well as slower release of bound nucleotide.

To conclude, the results presented here provide the blueprint of the inhibition versus activation of β-cardiac myosin by two different molecules targeting the same pocket. It illustrates how distinct restraints applied on the N-term, the Converter, and some connectors such as the Relay can govern distinct allosteric behavior in the myosin motor. Modulators with distinct scaffolds can bind and stabilize this pocket differently indicating how the plasticity would regulate drug affinity as well as allosteric effects in motor function. Interestingly, this pocket also affects myosin regulation by controlling the IHM formation. All-atom molecular dynamics grounded on high-resolution structures is of prime interest for understanding how force is generated and how myosin modulators can be conceived to regulate force generation in the context of human diseases. Cardiomyopathies can result from distinct mutations and the description of the OM/Mava pocket opens the door to exploring how to modify the potency of these derivatives for customizable effects on force production depending on the pathological phenotype to be corrected. Such a rational approach will also be of interest for other myosins since modulators are being studied to target other myosin-associated pathologies such as spasticity by targeting SkMyo2[47]; malaria by targeting myosin A[38,39,48], or asthma by targeting SmMyo2[49].

## Methods

### Protein purification

Bovine cardiac S1 fragment was purified from a frozen heart (Pel-Freez Biologicals) via the protocol described previously[29,34]. S1 was obtained by limited chymotryptic digestion of the full-length myosin in the buffer (20 mM K-Pipes, 10 mM K-EDTA, 1 mM EDTA, pH 6.8). The mix containing the protease (tosyl-Lysyl-chloromethane hydrochloride (TLCK)-treated from Sigma) was incubated at 22 °C for 30 min. The digestion was stopped by the addition of 1 mM phenylmethylsulfonyl fluoride (PMSF). Centrifugation at $29,000 \times g$ during 30 min at 4 °C allowed to removal of the insoluble myosin rods. The S1 fragment was precipitated with ammonium acetate (60% w/v final) and centrifugation ($29,000 \times g$ during 30 min at 4 °C). The pellet was resuspended and dialyzed with a low-salt buffer (12 mM K-Pipes, 2 mM MgCl₂, 1 mM DTT, 0.1 mM PMSF, pH 6.8). An anion-exchange chromatography step on Mono-Q (GE Healthcare) was performed at 4 °C in the buffer 20 mM Tris-HCl, 0.8 mM NaN₃, pH 8 with a 0-350 mM NaCl gradient. Fractions containing the purified S1 were pooled and buffer-exchanged in the buffer 10 mM HEPES, 50 mM NaCl, 1 mM NaN₃, 2.5 MgCl₂, 0.2 mM ATP, 1 mM TCEP, pH 7.5. S1 was finally concentrated at 42 mg.ml⁻¹ and 2 mM MgADP was added. The protein was aliquoted and flash-frozen in liquid nitrogen for storage.

### Mavacamten synthesis

Mava was synthesized in H.-J. Knölker's lab (see procedure below) and also purchased from Selleck Chemicals (www.selleckchem.com). The Mava powder was dissolved in DMSO at a stock concentration of 50 mM.

1-Isopropylbarbituric acid (1-isopropylpyrimidine-2,4,6(1H,3H,5H)-trione) (1)[50]

Dimethyl malonate (19.6 g, 148 mmol) and then sodium methanolate (18.2 g, 337 mmol) were added under continuous stirring to a solution of 1-isopropylurea (14.4 g, 141 mmol) in methanol (500 mL) at room temperature. The reaction mixture was heated under reflux for 18 h. Subsequently, the mixture was cooled first to room temperature, then to 0 °C, and acidified with hydrochloric acid (pH = 3). After the removal of the solvent in a vacuum, the residue was treated with ethanol (200 mL), and the resulting mixture was filtered. The filtrate was evaporated in a vacuum and the residue was purified by column chromatography on silica gel (dichloromethane/methanol, 20:1) to afford 1-isopropylbarbituric acid (13.9 g, 58%) as a light yellow oil.

6-Chloro-3-isopropylpyrimidine-2,4(1H,3H)-dione (2)[50]

Phosphorus oxychloride (37 mL) and benzyltriethylammonium chloride (26.1 g, 115 mmol) was added to 1-isopropylbarbituric acid (13.9 g, 81.7 mmol) under an argon atmosphere. The resulting mixture was heated at 50 °C for 18 h under an argon atmosphere. After cooling to room temperature, the excess phosphorus oxychloride was evaporated in a vacuum. The residual red oil was dissolved in dichloromethane (185 mL) and water (120 mL) was added slowly over a period of 1.5 h. The layers were separated, and the organic layer was washed with water (120 mL), and dried with sodium sulfate. After removal of the solvent, the residue was purified by column chromatography on silica gel (gradient elution with isohexane/ethyl acetate, from 5:1 to 1:1) to provide 6-chloro-3-isopropylpyrimidine-2,4(1H,3H)-dione (3.05 g, 20%) as a colorless solid.

(S)-3-Isopropyl-6-((1-phenylethyl)amino)pyrimidine-2,4(1H,3H)-dione (Mavacamten) (3)[50]

(S)-1-Phenylethan-1-amine (1.50 mL, 1.41 g, 11.6 mmol) was added to a solution of 6-chloro-3-isopropylpyrimidine-2,4(1H,3H)-dione (1.00 g, 5.30 mmol) in 1,4-dioxane (20 mL) and the resulting solution was heated at 80 °C for 24 h under an argon atmosphere. After removal of the solvent in vacuum, the residue was dissolved in ethyl acetate (70 mL), washed with 1 N HCl (2 × 50 mL), and then with a saturated aqueous solution of sodium chloride (40 mL). The organic layer was dried with sodium sulfate and the solution was concentrated in a vacuum to half of the original volume in order to induce the formation of a precipitation. Hexane (20 mL) was added and the mixture was stirred at room temperature for 10 min. The resulting solid was separated by filtration, washed with hexane (20 mL), and dried in a vacuum. The crude product was purified by column chromatography on silica gel (gradient elution with isohexane/ethyl acetate, from 1:1 to 1:3) to

afford (S)-3-isopropyl-6-((1-phenylethyl)amino)pyrimidine-2,4(1H,3H)-dione (439 mg, 30%) as a colorless solid.

The purity of the compound was checked by $^1$H NMR and GC-MS.

## Crystallization and data processing

The crystals studied in this work result from crystallization experiments that use a proteolytic S1 head preparation. In one crystal with Mava bound, the S1 fragment is crystallized and while in the other crystal, a motor domain (MD) fragment is present. These crystals are obtained by careful in situ proteolysis of the S1 head. The S1 contains the head region (1–710), with the Converter domain (711–780) and the first IQ region (781–810) complexed to the essential light chain (ELC). The MD contains the head region and the Converter. In crystallization solutions, ADP.Vanadate and ADP.BeF$_x$ is used to mimic hydrolyzed nucleotide (ADP.P$_i$) that stabilizes the pre-powerstroke state (PPS). Crystals of PPS-MD-Apo (type A) were obtained at 17 °C by the sitting drop vapor diffusion method from a 1:1 (v:v) ratio of protein (20 mg.ml$^{-1}$) with 2 mM Mg.ADP.Vanadate, trypsin at a ratio 1:1000 (w:w), and precipitant containing 0.5 M Lithium sulfate, 0.1 M Tris pH 8.5, and 25% PEG 3350. Crystals of PPS-S1-Mava (type B) were obtained at 4 °C by the hanging drop vapor diffusion method from a 1:1 (v:v) ratio of the protein (10 mg.ml$^{-1}$) with Mg.ADP.BeFx, 5 mM Mava-camten, trypsin at a ratio 1:500 (w:w), and precipitant 20.5% PEG3350, 7.5% NaTacsimate pH 6.0, 5 mM TCEP. Crystals of PPS-MD-Mava (type C) were obtained at 25 °C by the hanging drop vApor diffusion method from a 1:1 (v:v) ratio of the protein (10 mg.ml$^{-1}$) with 2 mM Mg.ADP.Vanadate, 0.5 mM Mavacamten, trypsin at a ratio 1:500 (w:w), and precipitant 23% PEG3350, 0.2 M Lithium sulfate, 0.1 M Tris pH 7.9. PPS-S1-OM (type D) was reprocessed from previous data, the crystallization conditions are documented in[29].

Crystals were transferred in the mother liquor supplemented with 30% glycerol and flash-frozen in liquid nitrogen. X-ray diffraction data were collected at the SOLEIL synchrotron (PX2A beamline, $\lambda = 0.98007$ Å and $\lambda = 0.9762$ Å for type A and D respectively; PX1 beamline, $\lambda = 0.98400$ Å and $\lambda = 0.97856$ Å for type B and C respectively), at 100 K. Diffraction data were processed using the XDS package[51] and AutoPROC[52] (version 2020). The reprocessing of data from crystal type D was performed with the same Rfree set as the one used in 5N69. Crystals type A belongs to the P4$_3$22 space group, crystals type 2 belongs to the space group P2$_1$, and crystals type C and D belong to the P2$_1$2$_1$2$_1$ space group; with one molecule per asymmetric unit for type A and C and two molecules per asymmetric unit. The data collection and refinement statistics are presented in Supplementary Tables 1 and 2.

## Structure determination and refinement

Molecular replacement was performed with bovine β-cardiac myosin (PDB code 5N69)[29] without water and ligand as a search model with Phaser[53] using: the motor domain (residues 80–710) for crystals type A and C and the entire S1 fragment for crystals type B and D. Iterative model building was performed using Coot[54] (version 0.9.2). Drug building and restrain libraries generations were achieved using ReadySet! from the phenix Suite[55,56] (version 1.19.2). Refinement was performed using Buster[57] (version 2020) with the highest resolution structure PPS-MD-Mava as a target structure. The statistics for most favored, allowed, and outlier in Ramachandran angles are for each crystal type respectively (in %): 95.64, 4.07 and 0.29 for PPS-Apo-MD; 97.58, 2.25 and 0.17 for PPS-S1-OM; 96.14, 3.30 and 0.57 for PPS-S1-Mava; 97.40, 2.45 and 0.14 for PPS-MD-Mava.

## Molecular dynamics

The molecular dynamics procedure is similar to[38,39], starting from the PPS-MD-Apo, PPS-S1-Mava, and PPS-S1-OM crystal structures. All molecular dynamics simulations were performed with Gromacs (version 2018.3)[58] on all-atom systems parametrized with charmm36m

forcefield[59] and built with CHARMM-GUI[60,61], that uses the following potential energy function (1) (see details in[62]):

$$V = \sum_{bonds} k_b(b-b_0)^2 + \sum_{angles} k_\theta(\theta - \theta_0)^2 + \sum_{dihedrals} [1 + \cos(n\varphi - \delta)]$$
$$+ \sum_{impropers} k_\omega(\omega - \omega_0)^2 + \sum_{Urey-Bradley} k_u(u - u_0)^2$$
$$+ \sum_{nonbonded} \varepsilon\left[\left(\frac{R_{minij}}{r_{ij}}\right)^{12} - \left(\frac{R_{minij}}{r_{ij}}\right)^6\right] + \frac{q_i q_j}{\varepsilon r_{ij}}$$

$$(1)$$

The Mg$^{2+}$, ADP, and Pi were built after Mg.ADP.Vanadate. All systems consisted of a box containing explicit waters TIP3 (97834 molecules), 150 mM KCl, and a pH set to 7.0 with no protonation of His residues. The box consisted of a cube of 149 Å as the length of the edge and a volume of 3307949 Å$^3$. Long-range interactions were handled using the particle mesh Ewald (PME) method[63] that allows to facilitation of the calculation of the energies. The Coulomb sum (E$^{(coul)}$) (2) is split into direct space sum (E$^{(dir)}$) (3) and reciprocal space sum (E$^{(rec)}$) (4) (see[64]) as follows:

$$E^{(coul)} = \frac{1}{2} \sum_n \sum_i \sum_{j \neq i} \frac{k_c q_i q_j}{|n.L + r_{ij}|} \qquad (2)$$

$$E^{(dir)} \simeq \frac{1}{2} \sum_i \sum_{j \neq i} \frac{k_c q_i q_j(1 - \text{erf}(\beta|r|))}{|r_{ij}|} \qquad (3)$$

$$E^{(rec)} = \frac{1}{2} \sum_n \sum_i \sum_{j \neq i} \frac{k_c q_i q_j \text{erf}(\beta|r|)}{|n.L + r_{ij}|} \qquad (4)$$

Where r$_{ij}$ is the interparticle distance, kc is Coulomb's constant and β is the Ewald coefficient. The electrostatic energy is considered as a periodic system expressed as a sum over all pairs of interactions. While the expression of Coulomb sum is conditionally convergent, the direct and reciprocal sums are rapidly convergent sums. The Ewald formulation is used in molecular simulations to express the electrostatic energy and force (see charmm-gui manual for details, https://www.charmm-gui.org/charmmdoc/ewald.html).

The simulations were performed in a constant temperature, constant pressure ensemble (NPT). The temperature and the pressure of the system were fixed at 310.15 K with the Nosé-Hoover thermostat and 1 bar with the Parrinello-Rahman barostat[65,66]. The total duration of the simulations was 720 ns for Apo; and 1020 ns for OM and Mava. The simulations were performed at least twice for each condition in order to ensure the reproducibility of the phenomena discussed in this work. Trajectories were generated and analyzed in PyMOL[67] (version 2.5.0) which served to create the illustrations. Atomic displacements were computed with VMD[68].

## Frame and statistical analysis

Trajectories were assembled with 'gmx trjconv' from Gromacs. Each trajectory was adjusted on the first frame by superimposing the atoms of the motor backbone using macros from VMD (RMSD trajectory tool, version 1.9.4a53, June 29, 2021), thus allowing the extraction of the RMSD plots. The first images were oriented in a previous alignment so that all representations are in the same referential.

The frames chosen to illustrate the conformation of the actin-binding interface (Fig. 5), the active site versus the position of the Lever arm, and the drug binding pockets (Supplementary Figs. 7, 9, and 10) were chosen for illustration purposes. Thus these illustrations are used to display the coupling between drug positions, priming of the Lever

arm, and the position of the backdoor but also the conformations of the actin-binding interfaces in the three conformations. Supplementary movies 1–7 display the full dynamics for all these regions.

The Geo-Measures plugin[69] (v. 0.9d) for PyMOL using MDTraj[70] and GROMACS[58] tool g_sham was used to generate the probability density function and the free energy surface. Two variables were used: the Root Mean Square Deviation (RMSD) on Cαs and the radius of gyration (Rg) on the entire S1 structure. As illustrated in Fig. 4, the Rg of the myosin increases when the Lever arm is unprimed since the conformation of the motor becomes more elongated. The plot RMSD function of the Rg is appropriate to monitor Lever arm swing. The Geo-Measures plugin was also used to monitor the orientation of the Lever arm with angles between the Cα of three residues. Two sets of residues were chosen (K103 < C705 > M659 and K803 < C705 > M90) to monitor the orientation of the Lever arm in two orientations during the simulation (Fig. 4). Interestingly, the shape of the curves of these angles is close to the shape of the curves of the RMSD (Fig. 4a). The trajectories were sampled at nanosecond intervals for all the analysis.

### Binding pockets analysis
The drug-binding pockets were analyzed manually. The plot representations in figures have been achieved using the software LigPlot+[71].

### Free energy calculations
The calculations of binding free energy (ΔGs) were performed with the software Prodigy[72,73] at 25 °C for all the frames of the time course of the simulations. The $K_D$ was derived from the Eq. (5).

$$\triangle Gs = RT\ln(K_D) \qquad (5)$$

$K_D$ values were then plotted as a function of time for the two conditions OM and Mava (Supplementary Fig. 11).

### Homology modeling
Homology modeling of O. cuniculus SkMyo2 PPS state binding OM has been performed with SwissModel[74] with the structure of PPS-S1-OM as a template. The figure with Mava was performed with the same model. The quality of the model has been checked by comparison with the available structure of SkMyo2 (PDB code 6YSY[47]).

### Reporting summary
Further information on research design is available in the Nature Portfolio Reporting Summary linked to this article.

## Data availability
The atomic models and the structure factors of PPS-MD-Apo, PPS-S1-Mava, PPS-MD-Mava, and PPS-S1-OM are available at the PDB under the accession numbers 8QYP, 8QYQ, 8QYR, and 8QYU respectively. Source data for all the graphs are provided. Source data are provided in this paper.

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

## Acknowledgements

The authors greatly acknowledge Dr. James Hartman (Cytokinetics Inc.) for providing purified β-cardiac myosin S1; Dr. Catherine Guillou and Dr. Thibault Bayles (Institut de Chimie des Substances Naturelles, ICSN, UPR2301, Gif-sur-Yvette, France) for providing the Mavacamten derivatives that helped identifying the initial crystallization conditions; the beamline scientists of PX1 and PX2A (SOLEIL synchrotron) for excellent support during data collection. The A.H. team is part of the Labex Cell(n) Scale ANR-11-LBX-0038 and IDEX PSL, which is part of the Initiatives of Excellence of Université Paris Sciences et Lettres (ANR-10-IDEX-0001-02-PSL). This work was supported by the CNRS, grants to A.H. from AFM 21805, FRM DCM20181039553, NIH RM1GM131981-01, and ANR-21-CE11-0022-01.

## Author contributions

A.H. designed and directed the research. J.R.P. and A.H. conceived and planned the experiments and were involved in project administration. G.T. performed the synthesis of Mavacamten. Protein crystallization was performed by J.R.P. with the help of A.D. Data collection and processing were performed by J.R.P. with the help of C.K., J.R.P., and C.K. performed model building and refinement. Molecular dynamics experiments were run by D.A. with the help of S.R. S.R. provided access to the calculators and helped to set up the calculations. A.W.S. and H.J.K. designed and supervised the chemical synthesis of Mavacamten and analyzed the synthesized compounds. Formal analysis and validation of the results was performed by D.A., J.R.P., and A.H. J.R.P., D.A., and A.H. wrote the manuscript with the help of S.R. and all authors reviewed it. A.H. provided funding.

## Competing interests

AH receives research funding from Cytokinetics and consults for Kainomyx. All other authors have no competing interests.
