## [Peer Review File · Nature Communications]

Reviewers' Comments:

Reviewer #1:

Remarks to the Author:

The present paper by Auguin et al. reports new X-ray crystallography based structures that visualize the binding site for mavacamten on cardiac beta-myosin II. This compound is a recently FDA-approved drug for treatment of obstructive hypertrophic cardiomyopathy. Interestingly, the authors find that mavacamten binds in the same site as omecamtiv mecarbil a myosin activator and a drug candidate in heart failure with reduced ejection fraction. By molecular dynamic simulations the authors also provide evidence that seems to explain several differences in the effects of the two drugs despite their binding in the same pocket.

The results are novel, and of interest, both for fundamental understanding of myosin function as a molecular motor and for future drug developments with myosin as a target. In the latter regard, the paper describes the residues which take part in binding of the different drugs. It also describes the different molecular effects of the two compounds and the basis for specificity for cardiac myosin II.

The paper is generally well and clearly written and the figures are mostly informative and clear, although in some cases a bit crowded. I have some suggestions in the latter regard (see below). I also have more extensive and important suggestions for improvements in the Introduction, Summary and Discussion where I wish to have some further clarifications related to clinical significance as follows:

-Summary, Line 18. This statement is not true. The prevalence of inherited cardiomyopathies is far lower than that of coronary artery disease and its consequences such as myocardial infarction, angina pectoris and heart failure. Heart failure also has several other reasons that are more common than inherited cardiomyopathies.

-Lines 33-34. The problem with the sentence on these lines is the same as mentioned in the above point.

-Beginning of Discussion. The statement on lines 390-392 is much too strong. Whereas myosin-active drugs may become important in treatment of cardiomyopathies (however, see Maron et al. JAHA 2022; DOI: 10.1161/JAHA.121.024566) their roles in treatment of heart failure are harder to see, particularly after OM was recently turned down by FDA (which may deserve to be mentioned). However, this does of course not mean that the effects of these drugs and their potential usefulness is not of interest. It is just the statement I have a problem with. Related, on line 395 "While such small-molecules will be used as treatments"; It is uncertain whether OM will be used as treatment in view of the recent FDA decision, although the EMA may have different views. Anyway, please consider modifying the sentence.

I am also somewhat doubtful about the use of the terms "antagonistic" and "antagonism", particularly in the title but also elsewhere. Pharmacologically, "antagonism" is usually taken to mean that one drug binds to a binding site without exerting an effect on its own on that site, instead inhibiting the binding of the normal agonist. An example is beta-blockers (antagonist without own effect) competitively inhibiting the binding of the normal agonist (activating the receptor) such as noradrenaline. In the present case, the obvious alternative view to "antagonism" is that there instead are two agonists with opposite effects when they bind to the same site. In this context, I think that use of the term "antagonistic" is ambiguous and could lead some readers astray in their thinking. The terminology issue needs to be carefully considered. The terms "positive allosteric modulator" or "negative allosteric modulator" may be useful in this context (cf. some recent papers that bring up the issue in different ways:

<https://doi.org/10.1371/journal.ppat.1008323>; <http://dx.doi.org/10.1124/pr.114.008862>; DOI 10.1002/ps.7699).

Whereas the study provides interesting and valuable information about structural aspects of the mechanisms of action of OM and mava, I had expected to find some information about to what extent the results elucidate the different effects of mava and OM on the Pi-release rate. The authors seem to suggest that mava slows Pi-release mainly by preventing the actin binding, which I presume means that the Pi-release state cannot form. Is that correct? Please clarify? On the other hand, the authors say nothing about the effects of OM to increase the Pi-release rate. Related, I also wonder if the authors could say something about the possible bearing of their

findings on different models for Pi-release in relation to the power-stroke (e.g. Mansson et al Bioessays 2023)?

Minor

Summary, Line 19. "late-stage" -> "late stage"

Summary, Line 20. "small-molecules" -> "small molecules"

Summary, Lines 21-23. It is true in principle that the compounds have gone through phase 3 clinical trial but it is of some interest to note that the outcomes were not the same. I suggest to add "with different outcomes" or something similar or simply remove this sentence.

Line 44. "myosin can indeed adopt" I suggest that you skip "indeed".

Line 45. "interacted-heads" -> "interacting heads"

Line 48. "small-molecules" -> "small molecules"

Line 51. Say something about different outcomes of clinical trials.

Line 55. "treatment, and is now available for patients of obstructive hypertrophic" -> "treatment, of moderately severe obstructive hypertrophic" Optional.

Line 64. "Fig. 1a, 1b). Although" -> "Fig. 1a, 1b). However,"

Lines 66-67. "The contribution of these modulators on the motor cycle.." -> "The effects of these modulators on the motor cycle, independent of their effect on the fraction of sequestered heads,.."
" I think that the suggested modification makes the message clearer provided that this is what you intend to say.

Lines 68-69. "..also time spent on actin.." -> "..also the time spent bound to actin.."

Line 70. "We solved the X-ray structure of OM-bound" -> "We previously (ref 25) solved the X-ray structure of OM-bound".

Otherwise, one gets the impression that this was first done in the present study whereas it is clear only from the next sentence that you refer to previous work.

Line 73. "Importantly, recent.." -> "Interestingly, recent.."

Otherwise, (if you prefer "importantly"), please clarify briefly already here why the different OM effects on wild-type and the R712L mutant are indeed important.

Line 100. "thus mandatory to dissect" -> "thus important to dissect" or "thus of great interest to dissect" or something similar.

Lines 104-105. "..the allosteric binding pocket of this inhibitor is also where OM, the activator of contraction, binds" -> "..the allosteric binding pocket of this inhibitor is the same as for OM"

Line 123. "..hydrolysis, yet slow Pi release in the absence of actin reducing.." -> "..hydrolysis, but slow Pi release in the absence of actin, reducing.."

Fig. 2. The figure is over-crowded. Please, increase space between second and last row of panels. Also increase distance between panels e and f. I would consider showing the figures in 4 rows instead of 3.

Legend Fig. 3, Line 199. "..Mava binding are colored in deep purple.." -> "..Mava binding are boxed in deep purple..?"

Line 226. "Lever arm up. The" -> "Lever arm up). The"

Legend Fig. 4 and related text.

i. Please motivate selections of different times in different panels, e.g. the times when most extreme angles were observed (if correct).

ii. One gets the impression of different average priming of the lever arms with mava and OM whereas it is stated above in the text that the average priming is the same. Please clarify.

iii. It is not clear from this figure (or elsewhere) that the simulations give similar results when repeated (as stated in the Supplementary information (Methods)).

Fig. 5. The font size is too small in the subscript in the inset in panel d.

Line 339. "..of two cycles,.." Do you mean "..of two cyclic groups.."

Line 398. "myosin diseases" is a strange term in my mind. I suggest "diseases where altered myosin function may play a role" or something similar.

Line 414. "In contrary" -> "In contrast"

SI

Legend Fig. S2. Line 46. It is the nucleotide position that is contoured in black with dashed lines.

Line 308. "charm-gui" -> "charmm-gui"

Line 310. "NPT ensemble" Please explain abbreviation for general reader.

Reviewer #2:

Remarks to the Author:

This report describes how the cardiac myosin inhibitor Mavacamten binds to the myosin head and compares it with the myosin activator Omecamtiv Mecarbil. The primary X-ray diffraction study is backed up by extensive molecular dynamics simulations, thus making for a very comprehensive description of the interactions of these small molecules with myosin from which mechanistic insights are deduced. The molecular dynamics measurements are largely qualitative and would be greatly improved by some more quantitative analysis. The primary finding is, as described in the title, that Mavacamten binds in the same pocket as the previously studied Omecamtiv Mecarbil, thus neatly demonstrating a principle that is becoming more apparent as studies accumulate, that the functional effects of small molecules depends very critically upon small differences in their structure. The predictions about the mechanism of Mavacamten and Omecamtiv Mecarbil function are plausible and should stimulate further experimentation, however it should be recognised that extrapolation from a single myosin head to intact myosin filaments is somewhat long-range and uncertain.

My criticisms are mostly confined to points of presentation which could improve the understanding of the data by those not wholly familiar with the subject.

1 lines 40-47. The introduction of the heart pathologies HCM and DCM is inadequate. It is necessary to clearly indicate the hypercontractile nature of HCM and the hypo-contractile phenotype of DCM (and heart failure in general-by definition). Otherwise the antagonistic effects of OM and Mava as therapies for the two phenotypes in the next paragraph do not make much sense.

2 Line 111-113 and supplement methods. Please define the molecules studied better. Whilst S-1 is a well known entity, the 'motor domain' is not. It is not defined (is it 1-780 ?) and the methods does not describe how it is made. Also please define PPS (prepowerstroke or postpowerstroke?) and give a brief explanation as to how the state is achieved by BeFx and/or Vanadate

3 Movies in supplement: Please label the movie files in the supplement properly. They are described as movie 1, 2 etc. but these do not correspond to the titles on the movies themselves (e.g. "463493_0_related_ms_8241587_s3bxk2" !). One cannot be sure one is looking at the right movie, which is critical since they provide such important information.

4 Three of the videos do not play on quicktime and one pdf does not display either. Can you rectify this?

463493_0_related_ms_8242804_s3bxk3.pdf

463493_0_video_8239850_s3bxjw.mp4

463493_0_video_8241582_s3bxjx.mp4

463493_0_video_8239849_s3bxjv.mp4

5 The MD studies could be improved with more quantitative analysis. For instance, instead of giving the extreme lever arm angles explored (Fig 4b) a plot of the probability vs angle would be hugely more informative, since it would show both the range of angles explored but also the degree of stability of the mean angle and even show the presence of biphasic or more complex distributions of the angle if they exist. Probability distribution plots of other parameters would also aid interpretation. In addition the probabilities of specific ligand atom to protein atom interactions could be calculated and presented to augment Figure 5 since this is really hard to follow.

6 Line 350 For the OM mechanism, can you take into account the main take-home message of Woody et al, that the lever arm barely moves during the 'power stroke' and this seems to be specific to OM (I.e. Mavacamten does not do it.) This property accounts for the cooperative potentiation effect of OM and the inhibitory action of OM at higher concentrations repeatedly seen in myofibrillar systems (i.e an effect like NEM-S1)

7 Lines 367 etc. It is a very good insight that the local effects of Mava and OM can effect the blocked-free head interface of two headed myosin, however the illustrations are hard to follow. Could you add space filling models like those used by Spudich lab that so elegantly illustrate the myosin mesa etc, (Biochemical Society Transactions (2015) Volume 43, part 1).

Reviewer #3:

Remarks to the Author:

This is an extremely detailed and rigorous presentation of new and surprising results of crystallographic studies of myosin, which make substantial progress toward understanding the mechanisms of action of two drugs in development. More data is needed, since all questions are not answered. The paper acknowledges potential ambiguities and does not pretend to answer all the questions, recognizing that the data have flaws (e.g., insufficient structural resolution to answer some key questions). However, this is an impressive study that makes significant progress toward mechanistic understanding needed to enable drug development in this field and others.

Manuscript submitted to Nature Communications - Rebuttal :

"Omecamtiv mecarbil and Mavacamten target the same myosin pocket despite antagonist effects in heart contraction".

REVIEWER COMMENTS *and rebuttal*

Reviewer #1 (Remarks to the Author):

The present paper by Auguin et al. reports new X-ray crystallography based structures that visualize the binding site for mavacamten on cardiac beta-myosin II. This compound is a recently FDA-approved drug for treatment of obstructive hypertrophic cardiomyopathy. Interestingly, the authors find that mavacamten binds in the same site as omecamtiv mecarbil a myosin activator and a drug candidate in heart failure with reduced ejection fraction. By molecular dynamic simulations the authors also provide evidence that seems to explain several differences in the effects of the two drugs despite their binding in the same pocket.

The results are novel, and of interest, both for fundamental understanding of myosin function as a molecular motor and for future drug developments with myosin as a target. In the latter regard, the paper describes the residues which take part in binding of the different drugs. It also describes the different molecular effects of the two compounds and the basis for specificity for cardiac myosin II.

The paper is generally well and clearly written and the figures are mostly informative and clear, although in some cases a bit crowded. I have some suggestions in the latter regard (see below). I also have more extensive and important suggestions for improvements in the Introduction, Summary and Discussion where I wish to have some further clarifications related to clinical significance as follows:

We thank Reviewer#1 for his/her positive comments. His comments improved significantly the quality of the paper. We have changed the introduction in order to better define the outcomes of the two molecules and be more accurate. We also changed the way we display molecular dynamics results are presented as requested by Reviewer #2.

-Summary, Line 18. This statement is not true. The prevalence of inherited cardiomyopathies is far lower than that of coronary artery disease and its consequences such as myocardial infarction, angina pectoris and heart failure. Heart failure also has several other reasons that are more common than inherited cardiomyopathies.

This sentence is corrected as follows:

"Inherited cardiomyopathies are common cardiac diseases worldwide"

-Lines 33-34. The problem with the sentence on these lines is the same as mentioned in the above point.

The sentence is corrected as follows: "Inherited cardiomyopathies are a global health concern, being a major cause of heart disease worldwide."

-Beginning of Discussion. The statement on lines 390-392 is much too strong. Whereas myosin-active drugs may become important in treatment of cardiomyopathies (however, see Maron et al. JAHA 2022; DOI: 10.1161/JAHA.121.024566) their roles in treatment of heart failure are harder to see, particularly after OM was recently turned down by FDA (which may deserve to be mentioned). However, this does of course not mean that the effects of these drugs and their potential usefulness is not of interest. It is just the statement I have a problem with. Related, on line 395 "While such small-molecules will be used as treatments"; It is uncertain whether OM will be used as treatment in view of the recent FDA decision, although the EMA may have different views. Anyway, please consider modifying the sentence.

We agree that the sentence was imprecise and we changed this sentence as follows: "Amongst these treatments, the activator *Omecamtiv mecarbil* OM is in phase 3 clinical trials against heart failure¹³ and *Mavacamten (Mava)* was approved by the FDA under the name of CAMZYOS™ to treat hypertrophic cardiomyopathies⁴³."

In the summary : Omecamtiv mecarbil and Mavacamten are two such molecules that completed phase 3 clinical trials, and the inhibitor Mavacamten is now approved by the FDA.

I am also somewhat doubtful about the use of the terms "antagonistic" and "antagonism", particularly in the title but also elsewhere. Pharmacologically, "antagonism" is usually taken to mean that one drug binds to a binding site without exerting an effect on its own on that site, instead inhibiting the binding of the normal agonist. An example is beta-blockers (antagonist without own effect) competitively inhibiting the binding of the normal agonist (activating the receptor) such as noradrenaline. In the present case, the obvious alternative view to "antagonism" is that there instead are two agonists with opposite effects when they bind to the same site. In this context, I think that use of the term "antagonistic" is ambiguous and could lead some readers astray in their thinking. The terminology issue needs to be carefully considered. The terms "positive allosteric modulator" or "negative allosteric modulator" may be useful in this context (cf. some recent papers that bring up the issue in different ways: <https://doi.org/10.1371/journal.ppat.1008323>; <http://dx.doi.org/10.1124/pr.114.008862>; DOI 10.1002/ps.7699).

To address the reviewer's concerns, we changed the title and the text as follows:

Title : "Omecamtiv mecarbil and Mavacamten target the same myosin pocket despite opposite effects in heart contraction"

Summary : All atoms molecular dynamics simulations reveal how these molecules produce distinct effects in motor allostery thus impacting force production in opposite way.

In the text : antagonistic is now replaced by opposite

Whereas the study provides interesting and valuable information about structural aspects of the mechanisms of action of OM and mava, I had expected to find some information about to what extent the results elucidate the different effects of mava and OM on the Pi-release rate. The authors seem to suggest that mava slows Pi-release mainly by preventing the actin binding, which I presume means that the Pi-release state cannot form. Is that correct? Please clarify?

We indeed observe that Mavacamten explores states that are distinct than those explored by OM and apo conditions, Exploration of these distinct states can slow P_i release due to two mechanisms: (1) the backdoor of the active site is closed at the beginning of the simulation preventing P_i release via the so-called P_i -release tunnel identified previously (Llinas et al 2015); (2) the actin interface explored by the states with mavacamten bound differ from those of wild type and OM and thus efficient actin binding that would promote the conformational changes required for P_i release are slowed.

The third paragraph of the Discussion was changed as follows:

“Detailed analysis coupled to molecular dynamics reveal how Mava triggers exploration of “incompetent” PPS states in which the actin interface and coupling within the motor are perturbed, consistent with the previously observed decrease of P_i release rate²⁴. In contrast, OM favors the exploration of structural states that are compatible with the opening of the back-door while keeping the Lever arm primed, states that are likely favoring inducing efficient binding to F-actin, as previously proposed from structures of the P_i release states of Myo6 (Llinas et al., 2015).”

On the other hand, the authors say nothing about the effects of OM to increase the Pi-release rate. Related, I also wonder if the authors could say something about the possible bearing of their findings on different models for Pi-release in relation to the power-stroke (e.g. Mansson et al Bioessays 2023) ?

Regarding OM, we find instead that the backdoor opens easily during the simulation and the actin binding interfaces explored are similar to those of wild type, thus suggesting that like apo, the actin interface is competent to bind quickly to the track and trigger P_i release. The difference with apo, is the fact that OM maintains the lever arm primed. It thus prevents the exploration of states that would correspond to reversing the recovery stroke.

This was already discussed in the Planelles et al 2017 paper, see legend of figure 5: “ATP hydrolysis stabilizes the Pre-powerstroke state, as does the binding of OM to the “PPS” site, which greatly slows the reversal of the recovery stroke (dotted pink lines)”. In other words, the rate of P_i release is likely increased since OM binding stabilizes the motor in a PPS primed state with ADP. P_i bound, making the number of motors ready to bind to the actin filament optimal. When no drug is present, some heads would be more easily found in intermediates of the recovery stroke with ATP bound, delaying P_i release compared to the condition in the presence of OM.

We have now clarified the discussion to indicate these points.

The edition of the third paragraph of the Discussion addresses thus these issues: “Detailed analysis coupled to molecular dynamics reveal how Mava triggers exploration of “incompetent” PPS states in which the actin interface and coupling within the motor are perturbed, consistent with the previously observed decrease of P_i release rate²⁴. In contrast, OM favors the exploration of structural states that are compatible with the opening of the back-door while keeping the Lever arm primed, states that are likely favoring efficient binding to F-actin inducing efficient P_i release, as previously proposed from structures of the P_i release states of Myo6 (Linas et al., 2015).” (Discussion, **third paragraph**).

Minor

Summary, Line 19. “late-stage” -> “late stage”.

This was corrected.

Summary, Line 20. “small-molecules” -> “small molecules”.

This was corrected. We also corrected in the other occurrences of this term.

Summary, Lines 21-23. It is true in principle that the compounds have gone through phase 3 clinical trial but it is of some interest to note that the outcomes were not the same. I suggest to add “with different outcomes” or something similar or simply remove this sentence.

This was corrected. “ *Omecamtiv mecarbil* and *Mavacamten* are two such molecules that completed phase 3 clinical trials, and the inhibitor *Mavacamten* is now approved by the FDA. In contrast to *Mavacamten*, *Omecamtiv mecarbil* acts as an activator of cardiac contractility.”.

Line 44. “myosin can indeed adopt” I suggest that you skip “indeed”.

This was corrected, “indeed” has been removed from the sentence.

Line 45. “interacted-heads” -> “interacting heads”

This was corrected.

Line 48. “small-molecules” -> “small molecules”

This was corrected.

Line 51. Say something about different outcomes of clinical trials.

This was corrected. “The most promising treatments against these diseases are small molecules directly modulating the force produced by β -cardiac myosin, the molecular motor driving heart contraction. *Omecamtiv mecarbil* and *Mavacamten* are two such molecules that completed phase 3 clinical trials, and the inhibitor *Mavacamten* is now approved by the FDA” (summary).

Line 55. “treatment, and is now available for patients of obstructive hypertrophic” -> treatment, of moderately severe obstructive hypertrophic” Optional.

We thank Reviewer#1 for this precision, it was corrected as follows: “Another modulator, the inhibitor *Mavacamten* (Mava), became the first revolutionary FDA-approved treatment, and is now available for patients of moderately severe obstructive hypertrophic cardiomyopathies (HCM) under the name of CAMZYOS™ (FDA, 2022)”.

Line 64. “Fig. 1a, 1b). Although” -> “Fig. 1a, 1b). However,”
This was corrected.

Lines 66-67. “The contribution of these modulators on the motor cycle..” -> “The effects of these modulators on the motor cycle, independent of their effect on the fraction of sequestered heads,..”
“I think that the suggested modification makes the message clearer provided that this is what you intend to say.

The sentence was changed: “The effects of these modulators on the motor cycle, independent of their effect on the fraction of sequestered heads, must also play an important role in their mechanism to modulate heart contraction”.

Lines 68-69. “..also time spent on actin..” -> “..also the time spent bound to actin..”
This was corrected.

Line 70. “We solved the X-ray structure of OM-bound” -> “We previously (ref 25) solved the X-ray structure of OM-bound”. Otherwise, one gets the impression that this was first done in the present study whereas it is clear only from the next sentence that you refer to previous work.
This was corrected.

Line 73. “Importantly, recent..” -> “Interestingly, recent...” Otherwise, (if you prefer “importantly”), please clarify briefly already here why the different OM effects on wild-type and the R712L mutant are indeed important.
This was corrected.

Line 100. “thus mandatory to dissect” -> “thus important to dissect” or “thus of great interest to dissect” or something similar.
This was changed. The sentence is now as follows: “It is thus essential to dissect the mode of action of OM and Mava on β -cardiac myosin, as this will guide the generation of novel drug candidates allowing all classes of mutations causing inherited cardiomyopathies to be treated”.

Lines 104-105. “..the allosteric binding pocket of this inhibitor is also where OM, the activator of contraction, binds” -> “.. the allosteric binding pocket of this inhibitor is the same as for OM”
This was corrected.

Line 123. “..hydrolysis, yet slow Pi release in the absence of actin reducing..” -> “..hydrolysis, but slow Pi release in the absence of actin, reducing..”
This was corrected.

Fig. 2. The figure is over-crowded. Please, increase space between second and last row of panels. Also increase distance between panels e and f. I would consider showing the figures in 4 rows instead of 3.

We increased the space between all the panels and reorganized the text in the figure. Now the figure appears less crowded.

Legend Fig. 3, Line 199. “..Mava binding are colored in deep purple..” -> “..Mava binding are boxed in deep purple..”?

This was corrected.

Line 226. “Lever arm up. The” -> “Lever arm up). The”

This was corrected.

Legend Fig. 4 and related text.

i. Please motivate selections of different times in different panels, e.g. the times when most extreme angles were observed (if correct).

We thank Reviewer #1 for this comment. To answer more precisely this request, we substantially edited Fig. 4 to include statistics. First, we have added two plots that follow the orientation of the Lever arm in two planes by measuring the angles defined using two couples of three residues during time: (i) for the first angle, K803 (tip of the Lever arm), C705 (base of the lever arm) and M659 (**Fig. 4b**) and (ii) for the second angle, K803, C705 and M90 were chosen (**Fig. 4c**). Interestingly, the evolution of these angles is similar to that of the RMSD (**Fig. 4a**) and it helps to follow more rigorously the angle of the converter in different planes. Moreover, we added two series of graphs to compare apo, Mavacamten and OM bound simulations (i) the RMSD function of the radius of gyration (Rg) of the entire protein is plotted with probability density functions of the conformations occurring during the simulation and (ii) the same graph but this time plotted with Gibbs free energy computed from the probability function (**Fig. 4g-i**). The Rg curve allows to see the unpriming of the lever arm, since the myosin is more elongated when the lever arm swings, leading to an increase of the Rg. These graphs also permits to better appreciate the differences between the conformations explored in the OM and Mava conditions, even if the lever arm is primed in the two conditions.

The text was also rewritten to fit the new **Fig. 4**:

“We monitored the Lever arm position by plotting the root-mean-square displacement (RMSD) for backbone C α atoms relative to their initial minimized complex structures during each simulation (C α RMSD plot, **Fig. 4a**). Additionally, we computed its orientation in two planes (**Fig. 4b-c**). In the Apo condition, the Lever arm swings back and forth, exploring conformations differing by up to $\sim 36^\circ$ (**Fig. 4a, Supplementary Fig. 7, 8 and Supplementary Movie 2**). All our indicators converged, as the RMSD curves follow a similar evolution than the orientations in the two planes (**Fig. 4a-c**). The Lever arm is the most mobile in the Apo condition, with large movements leading to explore less primed

Lever arm states. Mava slightly restrains the amplitude of the movements (**Supplementary Fig. 8, 9**). In contrast, OM binding significantly restrains the Lever arm movements and maintains it highly primed (**Fig. 4a-c; Supplementary Fig. 8, 9**). This could be demonstrated by statistical approaches such as expressing the RMSD function of the radius of gyration (Rg) of the entire protein, which can be plotted with density probability function of the frames (**Fig. 4d-f**) or with computed Gibbs free energy depending on the occurrence of the conformations during the simulation (**Fig. 4g-i**). The presence of OM and Mava both exert a cohesive action between the Lever arm and the motor domain that contributes to maintaining the Lever arm primed (**Supplementary Movie 2, 3 and 4**). Interestingly, OM and Mava significantly differ in their mobility within the pocket: OM remains for a longer duration in a specific position close to that found in the crystal structure, (**Supplementary Fig. 10, Supplementary Movie 5**). In contrast, Mava explores different positions in the pocket (**Supplementary Fig. 9, Supplementary Movie 6**). This likely explains the distinct Lever arm mobility observed in the two conditions (**Fig. 4d-i**) since the converter is part of the drug binding pocket. Thus, distinct fluctuations when Mava or OM is bound in the pocket correspond to distinct possible movements of the Converter/Lever arm orientation and position explored (**Supplementary Movie 7**).” (Second paragraph of the section “**Mobility of the primed Lever arm greatly differs when Mava or OM are bound**” in the Results).

Description of the procedure and description of the rationale for choosing some frames for illustration purpose has been added in **Material and methods**:

“ The Geo-Measures plugin for PyMOL (Kagami *et al.*, 2020) using MDTraj (McGibbon *et al.*, 2015) and GROMACS tool *g_sham* were used to generate the probability density function and the free energy surface. Two variables were used: the Root Mean Square Deviation (RMSD) on C α s and the radius of gyration (Rg) on the entire S1 structure. As illustrated in **Fig. 4**, the Rg of the myosin increases when the Lever arm is unprimed since the conformation of the motor becomes more elongated. The plot describing the RMSD function of the Rg is appropriate to monitor the Lever arm swing. The Geo-Measures plugin was also used to monitor the orientation of the Lever arm with angles between the C α of three residues. Two sets of residues were chosen (K103<C705>M659 and K803<C705>M90) to monitor the orientation of the lever arm in two orientations during the simulation (**Fig. 4**). Interestingly, the shape of the curves of these angles is close to the shape of the curves of the RMSD (**Fig. 4a**). The trajectories were sampled at nanosecond intervals for all the analysis. ” (Material and methods, section “**frame and statistical analysis**”).

Kagami LP, das Neves GM, Timmers LFSM, Caceres RA, Eifler-Lima VL, Geo-Measures: A PyMOL plugin for protein structure ensembles analysis, Computational Biology and Chemistry(2020) doi.org/10.1016/j.compbiolchem.2020.107322.

MDTraj: A Modern Open Library for the Analysis of Molecular Dynamics Trajectories. McGibbon RT, Beauchamp KA, Harrigan MP, Klein C, Swails JM, Hernández CX, Schwantes CR, Wang LP, Lane TJ, Pande VS. Biophys J. 2015 Oct 20;109(8):1528-32. doi: 10.1016/j.bpj.2015.08.015.

ii. One gets the impression of different average priming of the lever arms with mava and OM whereas it is stated above in the text that the average priming is the same. Please clarify.

We fully agree with this statement, we now added **Fig. 4b and 4c** that follows the orientation of the Lever arm in two planes during the time course of the simulation. It is more accurate and better support the statements of the text that were edited (see above).

iii. It is not clear from this figure (or elsewhere) that the simulations give similar results when repeated (as stated in the Supplementary information (Methods)).

The simulations were performed several times (see Methods) and what we are discussing in this paper was observed in all replicates. To strengthen our statements, we added statistical analysis (see **Fig. 4**) allowing to better illustrate the significance of the phenomenon described in the paper.

Fig. 5. The font size is too small in the subscript in the inset in panel d.

We changed the font size in all **Fig. 5** to increase the readability and the clarity.

Line 339. “..of two cycles,..” Do you mean “..of two cyclic groups..”

We thank Reviewer #1 for this proposition. This was corrected.

Line 398. “myosin diseases” is a strange term in my mind. I suggest “diseases where altered myosin function may play a role” or something similar.

This was corrected and now reads as “**myosin-related diseases**”

Line 414. “In contrary” -> “In contrast”.

This was corrected.

SI

Legend Fig. S2. Line 46. It is the nucleotide position that is contoured in black with dashed lines.

This was corrected.

Line 308. “charm-gui” -> “charmm-gui”.

This was corrected.

Line 310. “NPT ensemble” Please explain abbreviation for general reader.

This term is now defined : “**The simulations were performed in a constant temperature, constant pressure ensemble (NPT)**”.

Reviewer #2 (Remarks to the Author):

This report describes how the cardiac myosin inhibitor Mavacamten binds to the myosin head and compares it with the myosin activator Omecamtiv Mecarbil. The primary X-ray diffraction study

is backed up by extensive molecular dynamics simulations, thus making for a very comprehensive description of the interactions of these small molecules with myosin from which mechanistic insights are deduced. The molecular dynamics measurements are largely qualitative and would be greatly improved by some more quantitative analysis. The primary finding is, as described in the title, that Mavacamten binds in the same pocket as the previously studied Omecamtiv Mecarbil, thus neatly demonstrating a principle that is becoming more apparent as studies accumulate, that the functional effects of small molecules depends very critically upon small differences in their structure. The predictions about the mechanism of Mavacamten and Omecamtiv Mecarbil function are plausible and should stimulate further experimentation, however it should be recognised that extrapolation from a single myosin head to intact myosin filaments is somewhat long-range and uncertain.

My criticisms are mostly confined to points of presentation which could improve the understanding of the data by those not wholly familiar with the subject.

We thank Reviewer#2 for his/her positive appreciation of the manuscript. We also thank him/her for the comments that helped us to improve the readability of the paper for general audience.

1 lines 40-47. The introduction of the heart pathologies HCM and DCM is inadequate. It is necessary to clearly indicate the hypercontractile nature of HCM and the hypo-contractile phenotype of DCM (and heart failure in general-by definition). Otherwise the antagonistic effects of OM and Mava as therapies for the two phenotypes in the next paragraph do not make much sense.

We agree that detailing the differences between HCM and DCM would be useful. We then re-wrote the first paragraph of the introduction as follows:

“Inherited cardiomyopathies are a global health concern, being a major cause of heart disease worldwide. Hypertrophic cardiomyopathies (HCM) are characterized by hypercontractile cardiomyocytes leading over the years to fibrosis and ventricular hypertrophy¹. Dilated cardiomyopathies (DCM) are in contrast characterized by an hypocontractile phenotype^{2,3}. While the end-stage cardiomyopathies can all lead to sudden deaths^{4,5,6,7}, the contractile phenotype differs in HCM and DCM at a molecular level³. These diseases are associated with single point mutations of contractile proteins from the sarcomere, such as β -cardiac myosin⁸. The effect of some of these point mutations has been extensively studied, but remains poorly understood. Up to now, therapeutic approaches to treat end-stage inherited cardiomyopathy have been highly invasive, including cardioverter-defibrillator implantations and heart transplantation⁹.”

2 Line 111-113 and supplement methods. Please define the molecules studied better. Whilst S-1 is a well known entity, the ‘motor domain’ is not. It is not defined (is it 1-780 ?) and the methods does not describe how it is made. Also please define PPS (prepowerstroke or postpowerstroke?) and give a brief explanation as to how the state is achieved by BeFx and/or Vanadate

In order to help the reader and make it clear, some details were added to the material and methods in Supplementary information in the section “Crystallization and data processing” as follows:

“The crystals studied in this work result from crystallization experiments that use a proteolytic S1 head preparation. In one crystal with Mava bound, the S1 fragment is crystallized and while in the other crystal, a motor domain (MD) fragment is present. These crystals are obtained by careful *in situ* proteolysis of the S1 head. The S1 contains the head region (1-710), with the Converter domain (711-780) and the first IQ region (781-810) complexed to the essential light chain (ELC). The MD contains the head region and the Converter. In crystallization solutions, ADP.Vanadate and ADP.BeF_x are used to mimic hydrolyzed nucleotide (ADP.P_i) that stabilizes the pre-powerstroke state (PPS).”

Some indications about the regions crystallized were also added to the main text in the Results, in the section “**Mavacamten stabilizes the pre-stroke state**”:

“We successfully solved the structures of (i) the S1 fragment (3-807 in chain A and 3-798 in chain B) complexed with Mava and Mg.ADP.BeF_x (PPS-S1-Mava), (ii) the proteolyzed motor domain fragment (MD, 34-781) complexed with Mavacamten and Mg.ADP.BeF_x (PPS-MD-Mava) and (iii) the structure of the MD fragment (32-780) complexed to Mg.ADP.Vanadate (PPS-MD-Apo) at a resolution of 2.61; 1.80 and 2.76 Å respectively (**Supplementary Table 1&2**).”

3 Movies in supplement: Please label the movie files in the supplement properly. They are described as movie 1, 2 etc. but these do not correspond to the titles on the movies themselves (e.g. “463493_0_related_ms_8241587_s3bxk2” !). One cannot be sure one is looking at the right movie, which is critical since they provide such important information.

The Supplementary Movies were labelled properly during submission, unfortunately the system of access to the files is not under our control.

4 Three of the videos do not play on quicktime and one pdf does not display either. Can you rectify this?

463493_0_related_ms_8242804_s3bxk3.pdf
463493_0_video_8239850_s3bxjw.mp4
463493_0_video_8241582_s3bxjx.mp4
463493_0_video_8239849_s3bxjv.mp4

This was corrected. Now the movies display perfectly on a software such as windows media player. The .pdf also opens properly on adobe.

5 The MD studies could be improved with more quantitative analysis. For instance, instead of giving the extreme lever arm angles explored (Fig 4b) a plot of the probability vs angle would be hugely more informative, since it would show both the range of angles explored but also the

degree of stability of the mean angle and even show the presence of biphasic or more complex distributions of the angle if they exist. Probability distribution plots of other parameters would also aid interpretation. In addition the probabilities of specific ligand atom to protein atom interactions could be calculated and presented to augment Figure 5 since this is really hard to follow.

We thank reviewer #2 for this comment. We agree that the extreme angles shown were not accurate. We substantially edited Fig. 4 to include statistics in it. First, we have added the two plots that follow the orientation of the Lever arm in two planes, by measuring the angle between two couples of three residues during time: K803 (tip of the Lever arm), C705 (base of the lever arm) and M659 (**Fig. 4b**) and K803, C705 and M90 (**Fig. 4c**). Interestingly, the evolution of these curves is similar to the RMSD (**Fig. 4a**) and it helps to follow more rigorously the angle of the converter in different planes. Moreover, we added two series of graphs in the three conditions (i) the RMSD function of the radius of gyration (Rg) of the entire protein plotted with probability density functions of the conformations occurring during the simulation and (ii) the same graph but this time plotted with Gibbs free energy computed from the probability function (**Fig. 4g-i**). The Rg allows also to see the unpriming of the lever arm, since when the lever arm swings, the myosin is more elongated and there is an increase of the Rg. These graphs also permits to better appreciate the differences between the conformations explored in the OM and Mava conditions, even if the lever arm is primed in the two conditions.

The text was rewritten to fit the new **Fig. 4**:

We monitored the Lever arm position by plotting the root-mean-square displacement (RMSD) for backbone C α atoms relative to their initial minimized complex structures during each simulation (C α RMSD plot, **Fig. 4a**). Additionally, we computed its orientation in two planes (**Fig. 4b-c**). In the Apo condition, the Lever arm swings back and forth, exploring conformations differing by up to $\sim 36^\circ$ (**Fig. 4a, Supplementary Fig. 7, 8 and Supplementary Movie 2**). All our indicators converged, as the RMSD curves follow a similar evolution than the orientations in the two planes (**Fig. 4a-c**). The Lever arm is the most mobile in the Apo condition, with large movements leading to explore less primed Lever arm states. Mava slightly restrains the amplitude of the movements (**Supplementary Fig. 8, 9**). In contrast, OM binding significantly restrains the Lever arm movements and maintains it highly primed (**Fig. 4a-c; Supplementary Fig. 8, 9**). This could be demonstrated by statistical approaches such as expressing the RMSD function of the radius of gyration (Rg) of the entire protein, which can be plotted with density probability function of the frames (**Fig. 4d-f**) or with computed Gibbs free energy depending on the occurrence of the conformations during the simulation (**Fig. 4g-i**). The presence of OM and Mava both exert a cohesive action between the Lever arm and the motor domain that contributes to maintaining the Lever arm primed (**Supplementary Movie 2, 3 and 4**). Interestingly, OM and Mava significantly differ in their mobility within the pocket: OM remains for a longer duration in a specific position close to that found in the crystal

structure, (**Supplementary Fig. 10, Supplementary Movie 5**). In contrast, Mava explores different positions in the pocket (**Supplementary Fig. 9, Supplementary Movie 6**). This likely explains the distinct Lever arm mobility observed in the two conditions (**Fig. 4d-i**) since the converter is part of the drug binding pocket. Thus, distinct fluctuations when Mava or OM is bound in the pocket correspond to distinct possible movements of the Converter/Lever arm orientation and position explored (**Supplementary Movie 7**).” (Second paragraph of the section “**Mobility of the primed Lever arm greatly differs when Mava or OM are bound**” in the Results).

Description of the procedure has been added in **Material and methods**:

“ The Geo-Measures plugin for PyMOL (Kagami *et al.*, 2020) using MDTraj (McGibbon *et al.*, 2015) and GROMACS tool `g_sham` were used to generate the probability density function and the free energy surface. Two variables were used: the Root Mean Square Deviation (RMSD) on C α s and the radius of gyration (Rg) on the entire S1 structure. As illustrated in **Fig. 4**, the Rg of the myosin increases when the Lever arm is unprimed since the conformation of the motor becomes more elongated. The plot describing the RMSD function of the Rg is appropriate to monitor the Lever arm swing. The Geo-Measures plugin was also used to monitor the orientation of the Lever arm with angles between the C α of three residues. Two sets of residues were chosen (K103<C705>M659 and K803<C705>M90) to monitor the orientation of the lever arm in two orientations during the simulation (**Fig. 4**). Interestingly, the shape of the curves of these angles is close to the shape of the curves of the RMSD (**Fig. 4a**). The trajectories were sampled at nanosecond intervals for all the analysis. ” (Material and methods, section “**frame and statistical analysis**”).

Kagami LP, das Neves GM, Timmers LFSM, Caceres RA, Eifler-Lima VL, Geo-Measures: A PyMOL plugin for protein structure ensembles analysis, Computational Biology and Chemistry(2020) doi.org/10.1016/j.compbiolchem.2020.107322.

MDTraj: A Modern Open Library for the Analysis of Molecular Dynamics Trajectories. McGibbon RT, Beauchamp KA, Harrigan MP, Klein C, Swails JM, Hernández CX, Schwantes CR, Wang LP, Lane TJ, Pande VS. Biophys J. 2015 Oct 20;109(8):1528-32. doi: 10.1016/j.bpj.2015.08.015.

6 Line 350 For the OM mechanism, can you take into account the main take-home message of Woody *et al*, that the lever arm barely moves during the ‘power stroke’ and this seems to be specific to OM (i.e. Mavacamten does not do it.) This property accounts for the cooperative potentiation effect of OM and the inhibitory action of OM at higher concentrations repeatedly seen in myofibrillar systems (i.e an effect like NEM-S1)

We agree with the reviewer on this point. We have added in discussion the fact that our results are consistent with the findings of Woody *et al*.

“Single-molecule experiments demonstrated that OM is compatible with actin binding, although it maintains the Lever arm primed, acting as a suppressor of the powerstroke²⁶. Our results are fully consistent with this data, as the cohesive action of OM around its

binding site can help maintain the Lever arm primed with limited flexibility when it occupies the pocket. This can be explained by the elongated shape of OM that allows for example interactions of the compound with the Converter and the Relay, not possible for Mavacamten. These interactions stabilize the position of the lever arm. In contrast, Mava does not maintain the Lever arm in such a specific primed position, it slows actin binding due to the fact that the actin-binding interfaces of the motor explored when Mava is bound differ from those found when Mava is not bound since Mava uncouples the myosin subdomains.” (Third paragraph in the **discussion**).

We agree that adding this statement in discussion will increase its significance. It is indeed part of the take home message of this paper. The comparative study of OM and Mava extends what was previously described for OM in the Woody *et al.* article. It provides a direct mechanism for why OM inhibits the powerstroke.

7 Lines 367 etc. It is a very good insight that the local effects of Mava and OM can effect the blocked-free head interface of two headed myosin, however the illustrations are hard to follow. Could you add space filling models like those used by Spudich lab that so elegantly illustrate the myosin mesa etc, (Biochemical Society Transactions (2015) Volume 43, part 1).

We thank Reviewer #2 for his remark. In **Fig. 6** we aim to demonstrate that OM induces subtle differences making the binding of this compound incompatible with the head-head interface of the IHM. It is thus impossible to show that with space-filling models. However, we tried to improve the figure and added two panels (Fig. 6d-e) to make a zoom and better show the differences induced by the presence of OM and why Mava is compatible with the IHM. Moreover, the interface established with the BH is displayed as surface to help the reader understanding the figure.

A space-filling model has also been added in **Fig. 7**.

Reviewer #3 (Remarks to the Author):

This is an extremely detailed and rigorous presentation of new and surprising results of crystallographic studies of myosin, which make substantial progress toward understanding the mechanisms of action of two drugs in development. More data is needed, since all questions are not answered. The paper acknowledges potential ambiguities and does not pretend to answer all the questions, recognizing that the data have flaws (e.g., insufficient structural resolution to answer some key questions). However, this is an impressive study that makes significant progress toward mechanistic understanding needed to enable drug development in this field and others.

We thank Reviewer#3 for his/her positive comments. Some minor changes were introduced in the manuscript to bring a more quantitative analysis, and a better presentation of the molecular dynamics.